# Dual silencing of lipophagy and lipolysis in *Rhodnius prolixus* induces lipid droplet remodeling without TAG accumulation in the fat body

**Samara Santos-Araujo, Katia C. Gondim, Isabela Ramos**[ID]◦*

Instituto de Bioquímica Médica Leopoldo de Meis, Universidade Federal do Rio de Janeiro, Rio de Janeiro, Brazil

◦ These authors contributed equally to this work.
* isabela@bioqmed.ufrj.br

## Abstract

Autophagy and lipolysis are key metabolic pathways involved in lipid mobilization and energy homeostasis. In *Rhodnius prolixus*, a major vector of Chagas disease, previous studies have shown that silencing autophagy-related genes (*RpAtg8*, *RpAtg6*) or the brummer lipase gene *RpBmm* alters lipid metabolism and impairs flight activity. Here, we investigate the combined roles of lipolysis and lipophagy by performing a dual RNAi-mediated silencing of *RpAtg8* and *RpBmm* under both fed and starved conditions. In the fat body, dual silencing did not exacerbate triacylglycerol (TAG) accumulation beyond single knockdowns but led to significant enlargement of lipid droplets (LDs), suggesting adaptive remodeling. In contrast, the flight muscle of fed insects showed additive increases in TAG content, indicating synergistic impairment of lipid mobilization under this condition. Dual knockdown also modulated expression of lipid metabolism-related genes, including *AKHr*, *Dgat1/2*, and *Plin1*, in a tissue- and condition-specific manner. Combined silencing reduced vitellogenin levels in the hemolymph, significantly compromised oviposition, hatching success, survival, as well as flight capacity. Nonetheless, none of the phenotypes observed in the dual-silenced group showed additive effects when compared to the single knockdowns. Thus, while some phenotypes appeared additive, particularly in TAG accumulation in the flight muscle (only under fed conditions) and fat body LD remodeling, most impairments were not changed by the dual knockdown. These results highlight the crucial but apparent redundant roles of lipolysis and autophagy in maintaining lipid homeostasis and ensuring reproductive success in *R. prolixus*.

**Data availability statement:** All relevant data are within the manuscript and its Supporting information files.

**Funding:** This research was funded by the following grants: Cientista do Nosso Estado (CNE)- Fundação de Amparo à Pesquisa do Estado do Rio de Janeiro (FAPERJ) (www.faperj.br/), Instituto Nacional de Ciência e Tecnologia em Entomologia Molecular (INCT-EM)-Conselho Nacional de Desenvolvimento Científico e Tecnologico (CNPq) (http://cnpq.br/) and Coordenação de Aperfeicoamento de Pessoal de Nível Superior (CAPES) (www.capes.gov.br/) to I.R. and K.G. The funders had no role in study design, data collection and analysis, decision to publish, or preparation of the manuscript.

**Competing interests:** The authors have declared that no competing interests exist.

**Abbreviations:** Acc, acetyl-CoA carboxylase; AKHr, adipokinetic hormone receptor; ATGL, adipose triglyceride lipase; Atgs, autophagy – related genes; Bmm, brummer lipase; BSA, bovine serum albumin; CPT, carnitine palmitoyltransferase; Cq, cycle threshold; DAG, diacylglycerol; Dgat, diacylglycerol acyltransferase; DMSO, dimethyl sulfoxide; ECL, Enhanced Chemiluminescence; FFAs, free fatty acids; HSL, hormone sensitive lipase; LD, lipid droplet; PBS, phosphate buffered saline; Plin, perilipin; SDS−PAGE, SDS−polyacrylamide gel electrophoresis; Vg, vitellogenin; TAG, triacylglycerol; TBS, Tris-buffered saline.

## 1. Introduction

*Rhodnius prolixus*, a major vector of Chagas disease, is a strict hematophagous insect with highly specialized metabolic adaptations to process protein-rich blood meals (Lange et al., 2022). After feeding, digested lipids are absorbed by the midgut and transported via lipophorin, the major lipoprotein in the hemolymph, to the fat body, where they are stored predominantly as triacylglycerol (TAG) within lipid droplets (LDs) [1–3]. Additionally, the high influx of amino acids resulting from protein digestion is used in *de novo* lipid synthesis in the fat body [4,5]. *R. prolixus* can tolerate prolonged fasting, sometimes over 20 days in laboratory conditions, relying on lipid reserves for energy homeostasis through mitochondrial β-oxidation of mobilized TAG [2,6–8]. These characteristics make *R. prolixus* a particularly insightful model for studying insect lipid metabolism, and several aspects of lipid metabolic regulation have already been described in this species [3,5,7,9–15].

Canonically, TAG stored in LDs is mobilized during energy-demanding states primarily via neutral lipolysis, which liberates free fatty acids (FFAs) for mitochondrial β-oxidation [16–18]. This process involves a cascade of three enzymatic reactions: adipose triglyceride lipase (ATGL) initiates TAG breakdown into diacylglycerol (DAG) and FFA; hormone-sensitive lipase (HSL) hydrolyzes DAG into monoacylglycerol (MAG) and FFA; and MAG lipase releases the final FFA and glycerol [19]. ATGL, a member of the PNPLA2 family [20], plays a conserved role in lipid metabolism across organisms such as yeast, zebrafish, and mice, where its deficiency leads to TAG accumulation and varied physiological impairments [21].

In *Drosophila melanogaster*, the ATGL ortholog Brummer (Bmm) is expressed throughout development and is upregulated by fasting. Mutants lacking Bmm accumulate excess TAG, exhibit enhanced survival during starvation, as well as reduced locomotor activity [22–24]. In the brown planthopper *Nilaparvata lugens*, *Bmm* knockdown by RNAi disrupts reproductive output and increases resistance to starvation [25,26]. In *R. prolixus*, RNAi-mediated silencing of *Bmm* results in TAG accumulation in the fat body under starvation, enlarged LDs, and reduced ATP content. These insects display impaired locomotion and flight performance. When knockdown is performed prior to a blood meal, silenced females lay fewer, nonviable eggs, which exhibit collapsed morphology and low hatching success. These results demonstrate that adjusting Bmm abundance and activity is crucial for mobilizing lipid stores to meet the metabolic demands of both reproduction and starvation [27].

Autophagy, a conserved intracellular degradation mechanism, plays a vital role in recycling nutrients under nutrient-deprivation conditions [28]. More than 40 autophagy-related genes (Atgs) have been identified in yeast, many of which are conserved across species [4,20,29–31]. Among them, Atg8 (also known as LC3 in mammals) is a marker of autophagosomal membranes. It exists in a cytosolic form (LC3-I) and a membrane-bound, lipidated, form (LC3-II), essential for autophagosome maturation [32,33]. Selective autophagy variants, including mitophagy, ER-phagy, and lipophagy, target specific organelles via dedicated receptors [34,35]. Lipophagy refers specifically to the autophagic degradation of LDs, offering an alternative or complementary route to classical lipolysis for TAG mobilization [36–39].

Autophagy occurs constitutively in most cell types, maintaining organellar and proteomic integrity [40]. Its regulation is tightly linked to nutrient availability, through signaling pathways such as mTOR, AMPK, and PPARα, but can also be triggered by oxidative stress, hypoxia, or interactions with the unfolded protein response and ubiquitin-proteasome system [41–45]. Therefore, dissecting the function of autophagy under diverse nutritional and physiological contexts is essential to understanding its role in metabolic adaptation [46]. Previous studies from our group showed that, under starvation, *R. prolixus* silenced for *RpAtg8* accumulate TAG in both the fat body and flight muscle, exhibit reduced locomotor and flight performance, shortened lifespan, and altered expression of lipid metabolism-related genes [47]. Under fed conditions, phenotypes were milder but included impaired TAG mobilization to sustain forced flight activity [48].

Thus, taken together, data from insects silenced for *RpBmm* [27] and *RpAtg8* [47,48] support the notion that the interplay between autophagy, lipolysis, and lipid metabolism in *R. prolixus* is intricate and strongly influenced by the insect's nutritional status. Here, we investigated whether these pathways function redundantly or synergistically by simultaneously silencing *Atg8* (autophagy) and *Bmm* (lipolysis) under both fed and starved conditions. We found that in the fat body, simultaneous silencing did not increase TAG levels beyond those seen in single knockdowns, but it did lead to the formation of larger LDs, suggesting adaptive remodeling to support adequate lipid mobilization. In contrast, the flight muscle of fed insects exhibited additive TAG accumulation, indicating synergy between the two pathways. Moreover, the dual knockdown impaired vitellogenin synthesis, oviposition, hatching, and survival, although no additive effects were observed for reproductive parameters. Together, these results demonstrate that autophagy and lipolysis act in a coordinated, tissue- and condition-specific manner to maintain lipid homeostasis, and are crucial in sustaining reproduction and the high-energy physiological responses in *R. prolixus*.

## 2. Materials and methods

### 2.1. Gene identification

The sequences of *R. prolixus RpAtg8* (RPRC014434) and *RpBmm* (RPRC002097) were obtained from Vector Base (vectorbase.org) and were previously characterized by [49] and [27], respectively.

### 2.2. Insects

All females used in this work were obtained from our *insectarium*, where the insects were fed for the first time (as adult insects) with live-rabbit blood 14 days after the 5th instar nymph to adult ecdysis. After the first blood feeding, the insects (males and females) remained together to mate and generate the eggs that will provide the first-instar nymphs to maintain the *insectarium*. Mating is monitored by observing the female's oviposition rates and F1 eclosion rates during this first cycle of feeding. After this, all the adult females were fed every 21 days, and only fully gorged insects were used for the experiments (insects are allowed to feed at will and usually gain 6−7 times the initial body weight in 20−30 min). Thus, fully gorged mated females from the second or third blood feeding were used, and they were highly synchronized regarding blood feeding, digestion, and oviposition. The insects were maintained at 28 ± 2 °C, relative humidity of 65–85%, and a 12 h light/12 h dark cycle. All animal care and experimental protocols were approved by the institutional ethics committee (Ethics Committee on Animal Use – Federal University of Rio de Janeiro CEUA-UFRJ), process number 01200.001568/2013-87.

### 2.3. Extraction of RNA and cDNA synthesis

Adult female fat bodies, flight muscles and ovaries were dissected and homogenized in TRIzol reagent (Invitrogen, Carlsbad, CA, USA) for total RNA extraction. The integrity and quality of the RNA samples were analyzed via electrophoresis on a 2% agarose gel (UBS, Cleveland, OH, USA). The A260/A280 ratios of all the samples were between 1.8 and 2.2. Reverse transcription was carried out using a High-Capacity cDNA Reverse Transcription Kit (Applied Biosystems, Inc.,

Foster City, USA) with 1 µg of total RNA after RNase-free DNase I (Thermo Fisher Scientific, Waltham, USA) treatment, according to the manufacturer's protocol.

## 2.4. PCR/quantitative PCR (qPCR)

PCR and qPCR experiments were performed using the Taq DNA polymerase enzyme (Thermo Fisher Scientific, USA) and specific primers designed to target *RpAtg8* and *RpBmm* as previously described [27,49]. Briefly, the cycle parameters were as follows: 5 min at 95 °C; 35 cycles of 30 s at 95 °C, 30 s at 50 °C and 1 min at 72 °C; and 15 min at 72 °C. For qPCR, the reactions were performed using the qPCRBIO SyGreen Mix Separate-ROX Kit (PCR Biosystems Ltd., London, UK). Processing was performed in a StepOnePlusTM thermal cycler (Applied Biosystems, Foster City, USA). The program was 95 °C for 10 min; 95 °C for 15 s; 60 °C for 45 s for 40 cycles; and a dissociation curve. For each sample, the cDNA was diluted 10 times. The Cq values obtained for the blanks were at least ten units above the experimental points. The blanks were created by replacing the amount of cDNA with Milli-Q water. *Rp18S* gene amplification was used for normalization, as previously described [50], and its amplification was constant under our experimental conditions, confirming that it was an appropriate endogenous control [51]. The relative expression and ΔΔCq values were calculated from the obtained cycle threshold (Cq) values [52]. The same cDNAs were used to verify the gene expression levels of different genes involved in different pathways of lipid metabolism measured by qPCR as described above and using primers previously described [47,48]. All primers are listed in S1 Table.

## 2.5. RNAi knockdown

Double-stranded RNAs (dsRNAs) for the genes *RpAtg8* and *RpBmm* were synthesized using the MEGAScript RNAi kit (Ambion, Inc., Austin, USA) with primers previously described [27,49]. The unrelated bacterial gene *MalE* (Gene ID: 948538) was used as a control dsRNA [53]. One microgram of each dsRNA (dsAtg8, dsBmm, dsAtg8 + dsBmm, or dsMal) was injected into the hemocoel of adult females using a 10 µL syringe (Hamilton Company, Reno, USA) following two different protocols depending on the condition studied. Protocol 1 (fed state): the females were injected 18 days after a blood meal, fed 3 days later, and dissected on days 5 and 10 after the blood meal; Protocol 2 (starving state): Females were fed and injected 10 days after the blood meal, and dissected on day 24 post-feeding (Fig 1). Subsequently, qPCR quantifications were performed on days 5, 10, and 24 post-feeding. Knockdown efficiency was verified in each experiment. All primers are listed in S1 Table.

## 2.6. Determination of TAG and protein content

The total abdominal ventral fat body, all flight muscles in the thorax and ovary were dissected from control and knockdown insects on the 5th, 10th and 24th days after the blood meal. The organs were washed in cold PBS buffer (10 mM sodium phosphate buffer, pH 7.4, 0.15 M NaCl) and individually homogenized in a Potter–Elvehjem tube in 200 µL of cold PBS for the flight muscle and ovary, and 150 µL for the fat body. The TAG content was determined enzymatically using the Triglycerides 120 kit (Doles Reagents, Goiânia, Brazil). TAG quantification was expressed as total TAG per organ, since the irregular and lobulated structure of *R. prolixus* fat bodies makes wet mass measurements unreliable due to hemolymph retention. Normalization by protein content was avoided, as it may vary with gene silencing and mask effects on TAG. This approach, of reporting TAG per organ, has been adopted in previous studies [2,5,12] and is consistent with recommendations by [54]. The total protein content was determined according to the methods of [55] using bovine serum albumin (BSA) as a standard.

## 2.7. Nile red staining of lipid droplets (LDs)

Fat bodies were collected from dsRNA-treated females (at least 3 females per condition) on days 5, 10, and 24 post-feeding and stained with Nile Red (Sigma-Aldrich, Saint Louis, MO, USA), as previously described for *R. prolixus* lipid droplet (LD) analysis [56]. The fat bodies were incubated for 15 minutes in 1 mg/mL Nile Red prepared in 75% glycerol.

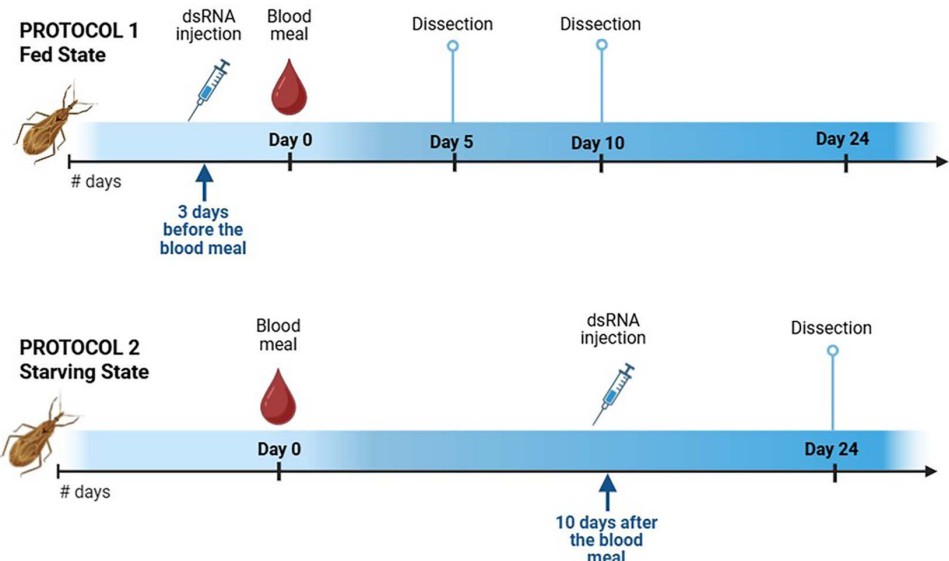

**Fig 1. Protocols used for gene silencing by RNA interference in the fed and starving states.** Protocol 1. Females were injected on day 18 after feeding, fed three days later and dissected on days 5 and 10 (fed state). Protocol 2. Females were injected on day 10 after feeding and dissected 14 days after injection, i.e., on day 24 after the blood meal (starving state).

The tissues were mounted on slides in 100% glycerol and immediately imaged using a Leica TCS-SPE laser scanning confocal microscope equipped with a 20×/0.70 NA Plan-Apochromatic objective lens (Leica Microsystems, LAS X software). Excitation wavelength used was 543 nm for Nile Red.

Peripheral regions of the fat body were selected for imaging because they exhibit reduced tissue thickness and allow for a standardized area of observation across organs from different insects. For each insect, two randomly chosen fields from the periphery of different lobes were acquired and analyzed. Each field represented approximately 4–6% of the total tissue area, corresponding to approximately 10% of the tissue analyzed per animal. Three independent experiments were conducted. The diameters of lipid droplets (LDs) were measured from two images per group using the image analysis software DAIME, following automatic edge detection–based segmentation [12,57]. The segmentation parameters applied in DAIME were: Edge detection – inclusion of dark regions: *no*; dark threshold: *140*; General options – ignore objects up to: *4*. Representative segmented images used for quantification are provided in S1 Fig. The LD diameters were plotted in frequency histograms.

## 2.8. Oviposition, hatching, survival, and digestion

Females injected with dsRNA were separated into individual tubes after feeding, and their oviposition was monitored daily until the end of the laying cycle. The eggs laid were also separated by group into individual tubes at the end of each oviposition day to monitor hatching. Regarding survival, the silenced females were also monitored daily until all had died. For the digestion experiment, dsRNA-injected females were dissected on days 1, 5, and 10 after the blood meal. The midguts (anterior midgut and posterior midgut, including luminal contents) were dissected and individually homogenized in 500 μL of PBS using a Potter-Elvehjem tube for total protein content determination.

## 2.9. Protein profile of the hemolymph

Females injected with dsRNA using the Protocol 1 (fed state) had their hemolymph collected on day 5 after the blood meal using a micropipette. The collected material was immediately mixed with phenylthiourea and centrifuged for 5 minutes

at 3000 rpm. The supernatant was then collected and frozen at −20 °C. Total protein concentration was determined as described above. Protein composition was analyzed by sodium dodecyl sulfate-polyacrylamide gel electrophoresis (SDS-PAGE, 7.5%). An egg sample was used to confirm the identification of vitellogenin. After electrophoresis, gels were stained with 0.3% (w/v) colloidal Coomassie Blue; 9% (v/v) glacial acetic acid; and 46% (v/v) methanol for 1 hour, followed by destaining with a solution of 5% (v/v) methanol and 7.5% (v/v) acetic acid. Finally, densitometry of the bands on the gel was performed using TotalLab Quant v11 software (TotalLab Ltd, Newcastle, United Kingdom).

## 2.10. Anti-Atg8 immunoblotting

Control and knockdown females were dissected on days 10 and 24 after the blood meal. Fat bodies (pools of three organs) were homogenized using a Potter-Elvehjem tube in 100 µL of cold PBS, while flight muscles were homogenized individually in 200 µL of PBS. To assess RpAtg8 protein levels in control and knockdown groups, 60 µg of total protein was used. All samples were separated by 13% SDS-PAGE, and proteins were subsequently transferred to nitrocellulose membranes (GE Healthcare Life Sciences, USA). Membranes were blocked for 1 hour in blocking buffer (10 mM Tris, pH 7.5; 0.15 M NaCl; 0.1% Tween-20; and 5% skim milk) and incubated overnight at 4 °C with rabbit-raised primary antibodies (1:2,500 in blocking buffer) against *R. prolixus* RpAtg8 [49]. Membranes were washed 3 times for 10 minutes with blocking buffer and then incubated for 1 hour with secondary antibody (goat anti-rabbit horseradish peroxidase [HRP] conjugate, Ab6721; Abcam, Cambridge, MA, USA) diluted 1:20,000. Two different primary antibodies were used as loading controls. Polyclonal rabbit anti-tubulin (#2144 Cell Signaling Technology, Danvers, MA, USA) at 1:5,000 was used for fat body samples, and anti-β-actin (Santa Cruz Biotechnology, Santa Cruz, CA, USA) at 1:1,000 was used for flight muscle samples; both were incubated overnight at 4 °C. For tubulin, the secondary antibody used was goat anti-rabbit HRP (Ab6721, Abcam) diluted 1:20,000 for 1 hour incubation. For β-actin, the same HRP-conjugated goat anti-mouse secondary antibody diluted 1:3,000 (Ab6789, Abcam) was used with the same incubation time. After washing membranes with blocking buffer, they were developed with an enhanced chemiluminescence system (ECL) (2.5 mM luminol in dimethyl sulfoxide [DMSO], 0.4 mM coumaric acid, 0.02% hydrogen peroxide in water, and 0.02% 1 M Tris, pH 8.4) for 1 minute. Band intensities were analyzed by densitometry using ImageJ software version 1.50i (NIH Image, Bethesda, MD, USA) with background corrections. The original uncropped images of all immunoblottings are shown in S2 Fig.

## 2.11. Assay for forced flight activity

Females treated with dsRNA were subjected to a forced flight test on days 10 and 24 after the blood meal [58,59]. Briefly, the insects were tethered by a thread attached to the dorsal surface of the thorax. A fan generated a continuous airflow, and the insects flew until exhaustion. Insects were considered exhausted when they stopped flying for more than 30 seconds despite continuous stimulation by the airflow. Flight duration was recorded for each insect.

## 2.12. Statistics

The ΔΔCq values were calculated from the obtained cycle threshold (Cq) values and were used for statistical analyses. Relative expression values ($2^{-\Delta\Delta Cq}$) were used only for data plotting. Results were analyzed using Student's t-test to compare two different conditions and one-way ANOVA followed by Tukey's test to compare more than two conditions. Oviposition and digestion results were analyzed by two-way ANOVA followed by Tukey's test. To compare LD diameters, the Kruskal-Wallis test followed by Dunn's test was used. Hatching was analyzed by the $\chi^2$ test, and survival by the Log-rank (Mantel-Cox) test. Differences were considered significant at $p < 0.05$, and all statistical analyses were performed using Prism 8.0 software (GraphPad Software, San Diego, CA, USA).

# 3. Results

## 3.1. Dual silencing of *RpAtg8* and *RpBmm* in the fat body does not produce additive effects on TAG or protein accumulation

Specific double-stranded RNAs targeting *RpAtg8* and *RpBmm* of *R. prolixus* were synthesized and injected into adult females following two experimental protocols to assess their effects under distinct nutritional conditions. In Protocol 1 (fed state), females were injected 18 days post-blood meal, fed 3 days later, and dissected either 5 or 10 days after feeding (Fig 1). In Protocol 2 (starving state), injections occurred 10 days after the blood meal, and dissections were performed 14 days later (Fig 1).

We confirmed efficient mRNA silencing for both targets, with *RpAtg8* reduced by at least 75% and *RpBmm* by approximately 40% in the fat body, whether silenced individually or simultaneously (Fig 2A, 2B). To investigate whether *RpAtg8* knockdown translated into reduced protein levels, we performed immunoblotting followed by densitometric analysis. The results showed a significant reduction in total *RpAtg8* protein levels, including both free (Atg8-I) and lipidated (Atg8-II) forms, at both day 10 and day 24 after the blood meal (Fig 2C, 2D). Silencing of *RpBmm* alone did not alter *RpAtg8* protein levels in the fat body at either time point.

To investigate the phenotypic effects of simultaneously inhibiting two major lipid degradation pathways, we quantified TAG content across the four experimental groups. Given that autophagy mediates the degradation of cytosolic components, including proteins [60], we hypothesized that inhibition of autophagy-related genes might also impact total protein levels in the fat body. Despite high variability among biological replicates, a general trend toward accumulation of both TAG and total protein was observed following individual and combined silencing of *RpAtg8* and *RpBmm*. Statistically significant differences were detected for protein levels on day 5 and for TAG levels on day 24 (Fig 3). Interestingly, no evidence of additive effects was found in the double knockdown condition, suggesting that, under these experimental conditions, the two pathways do not act in a complimentary or compensatory manner in mobilizing TAG reserves in the fat body, either in the fed or starved state.

## 3.2. Combined silencing of *RpAtg8* and *RpBmm* in the flight muscle leads to additive TAG accumulation in the fed state

In the flight muscle, mRNA silencing was efficient for both targets, with *RpAtg8* expression reduced by at least 90% and *RpBmm* by approximately 60%, whether silenced individually or in combination (Fig 4A, 4B). Immunoblotting and densitometric analysis confirmed a significant reduction in total RpAtg8 protein levels, including both the free (Atg8-I) and lipidated (Atg8-II) forms, at both day 10 and day 24 after the blood meal (Fig 4C, 4D). Silencing of *RpBmm* alone did not affect *RpAtg8* protein levels at either time point. With respect to energy reserves, we observed an additive accumulation of TAG in the dual knockdown condition during the fed state (day 5), relative to the single gene knockdowns (Fig 5). These findings suggest that, in the flight muscle during the fed state, lipolysis and lipophagy may partially compensate for each other in mobilizing TAG stores.

## 3.3. Silencing of *RpAtg8* and *RpBmm* modulates lipid metabolism gene expression in a tissue- and nutritional status-specific manner

To evaluate whether *RpAtg8* and *RpBmm* silencing affected the expression of other genes involved in lipid metabolism, we quantified the transcript levels of key lipid metabolic regulators in the fat body and flight muscle. The selected genes included adipokinetic hormone receptor (*AKHr*), acetyl-CoA carboxylase (*Acc*), carnitine palmitoyltransferase 1 (*Cpt1*), diacylglycerol acyltransferases 1 and 2 (*Dgat1* and *Dgat2*), and Perilipin 1 (*Plin1*).

In the fat body, individual *RpAtg8* silencing led to a significant 50% downregulation of *Dgat2* on day 5 and a trend towards a 5-fold upregulation of the same enzyme on day 24 (Fig 6A and S3 Fig). Silencing of *RpBmm* alone resulted in

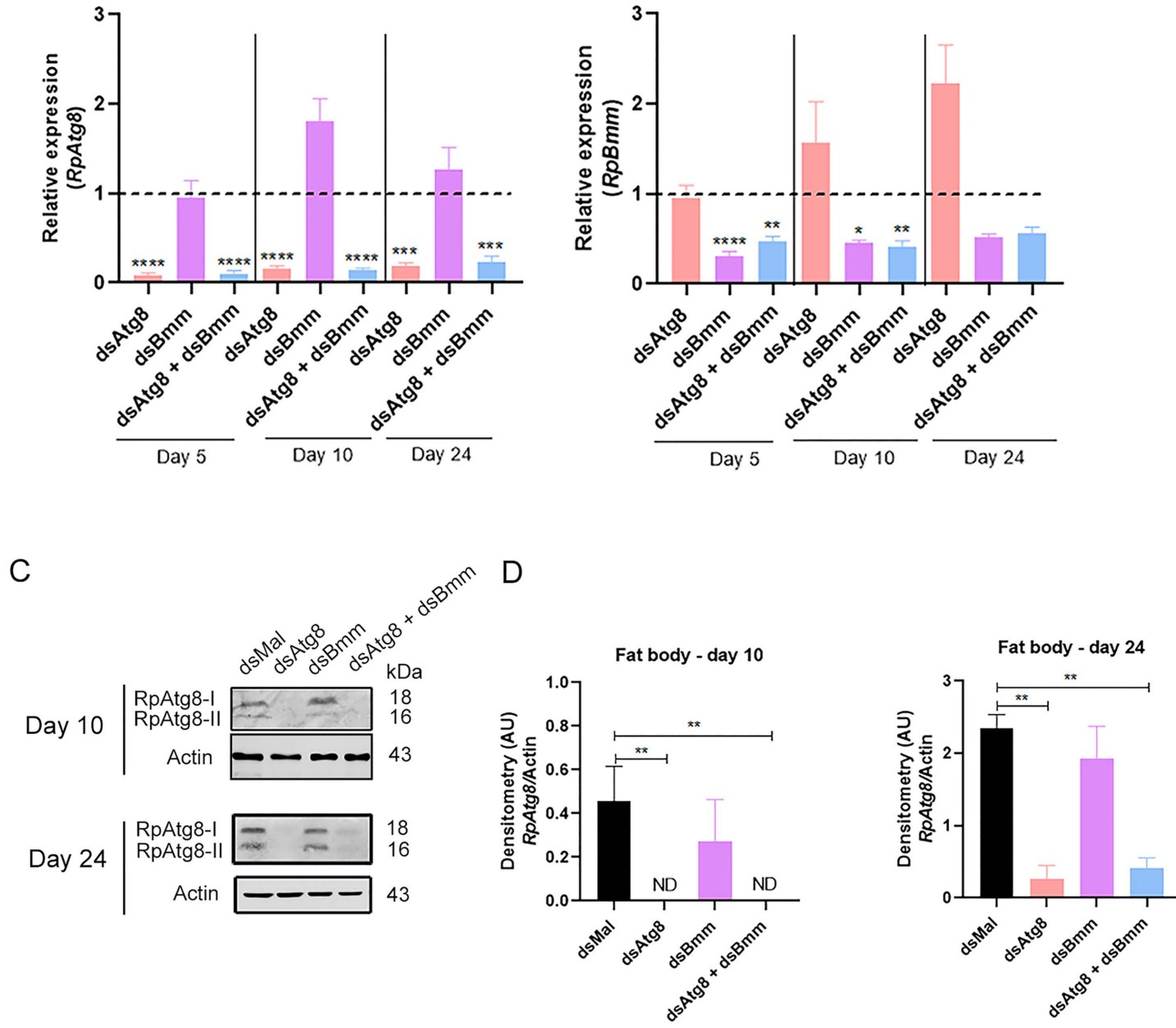

**Fig 2. Silencing efficiencies of *RpAtg8* and *RpBmm* in the fat body.** Adult females, 18 days after a blood meal, were injected with 1 µg of dsRNA for *RpAtg8*, *RpBmm*, *RpAtg8* + *RpBmm*, or *Mal* (control), fed three days later, and dissected either five or ten days after feeding (Protocol 1) or injected on the tenth day after feeding and dissected 14 days after injection (Protocol 2) (n = 4-7). **(A–B)** mRNA levels were determined by qPCR, using *Rp18S* expression as a reference gene. *RpAtg8* and *RpBmm* mRNA quantification in the fat body. Gene expression levels are relative to each control value (dashed lines). **(C)** RpAtg8 immunoblotting in the fat body on the 10th day (top image) and the 24th day (bottom image) (n = 3). **(D)** Total RpAtg8 densitometry on the 10th and 24th days. RpAtg8-I: free RpAtg8. RpAtg8-II: lipidated RpAtg8. All graphs show mean ± SEM. *p < 0.05, **p < 0.01, ***p < 0.001, ****p < 0.0001, when compared by one-way ANOVA followed by Tukey's post-test.

# Fat Body

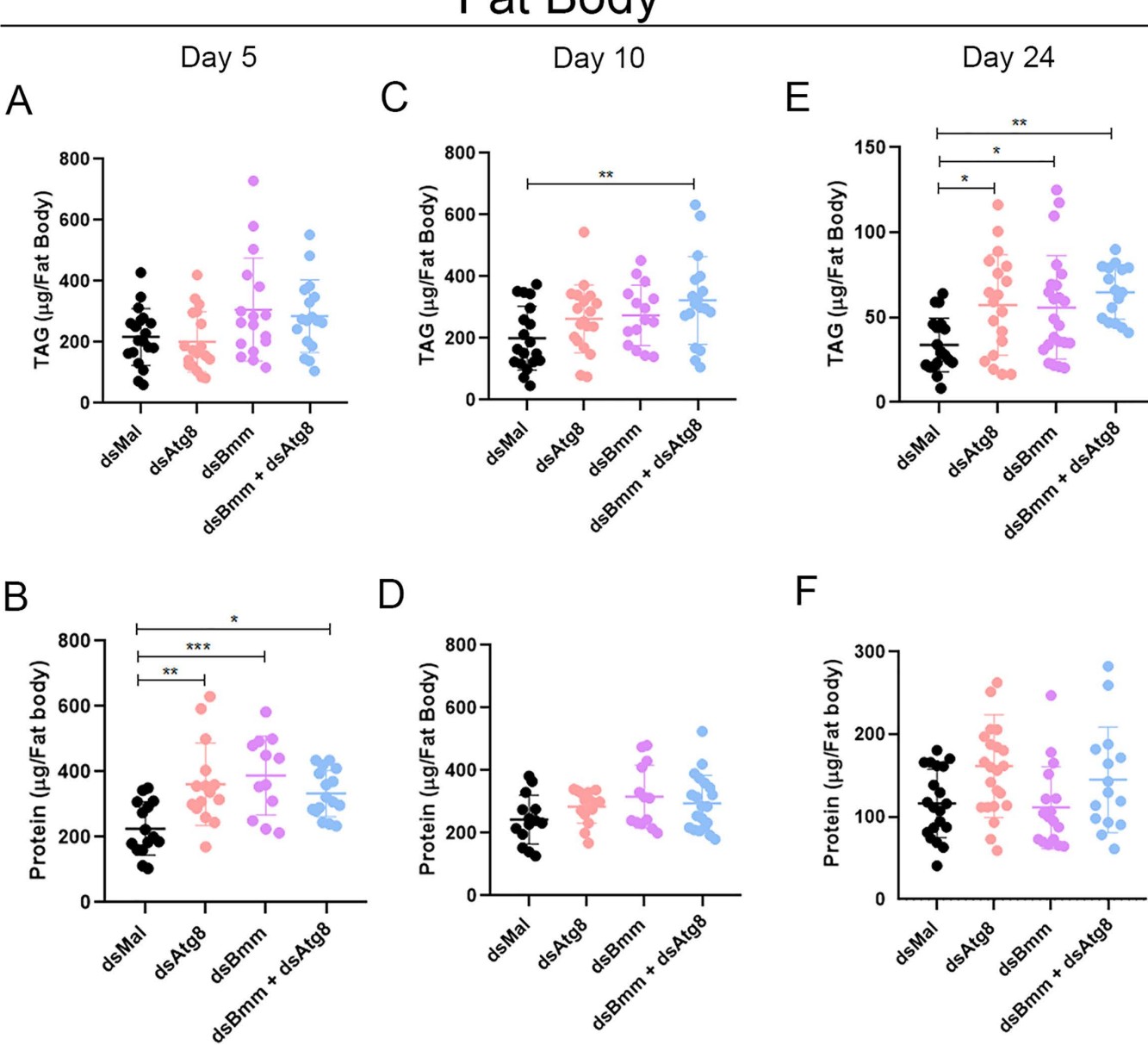

**Fig 3. Silenced females accumulate TAG in the fat body during starvation.** Adult females (18 days after a blood meal) were injected with 1 μg of dsRNA for *RpAtg8*, *RpBmm*, *RpAtg8+RpBmm*, or *Mal* (control), fed three days later, and dissected either five or ten days after feeding (Protocol 1) or injected on the tenth day after feeding and dissected 14 days after injection (Protocol 2). Fat bodies were collected, washed, individually homogenized, and total TAG **(A, C, E)** and protein content **(B, D, F)** on the 5th (A, B), 10th (C, D), and 24th (E, F) day after the blood meal were determined. Graphs show mean ± SD (n = 12–23 insects, obtained from 3 independent experiments). *p < 0.05, **p < 0.01, ***p < 0.001, when compared by one-way ANOVA followed by Tukey's post-test.

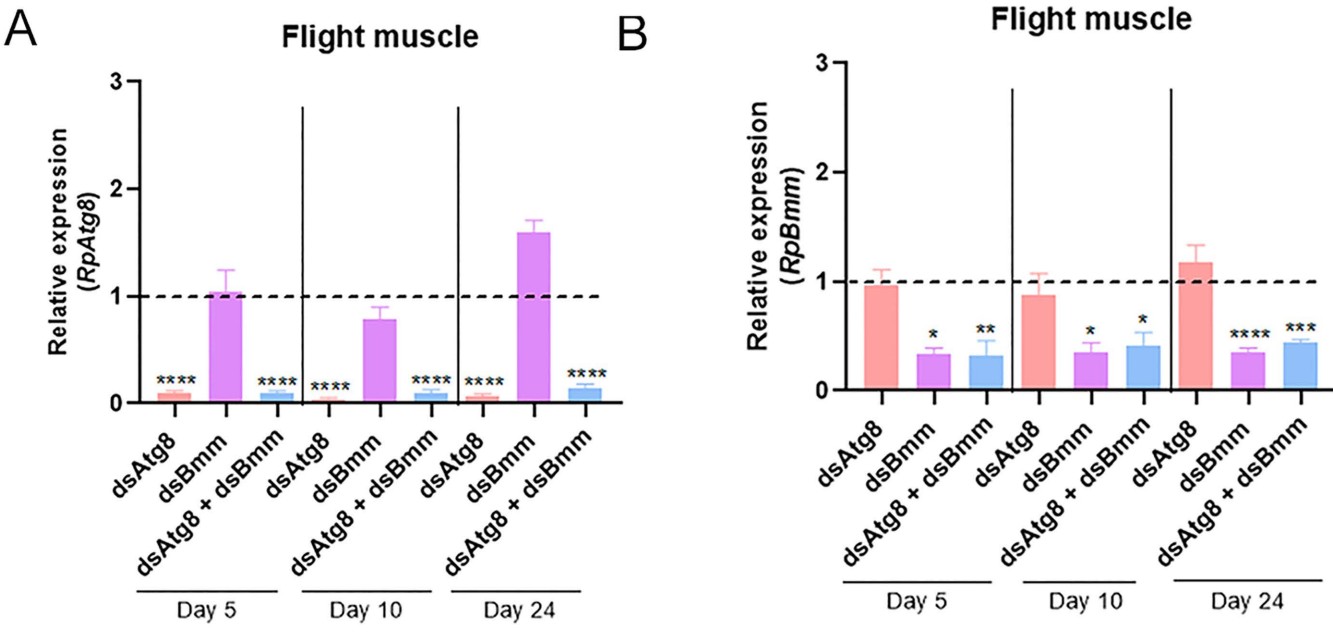

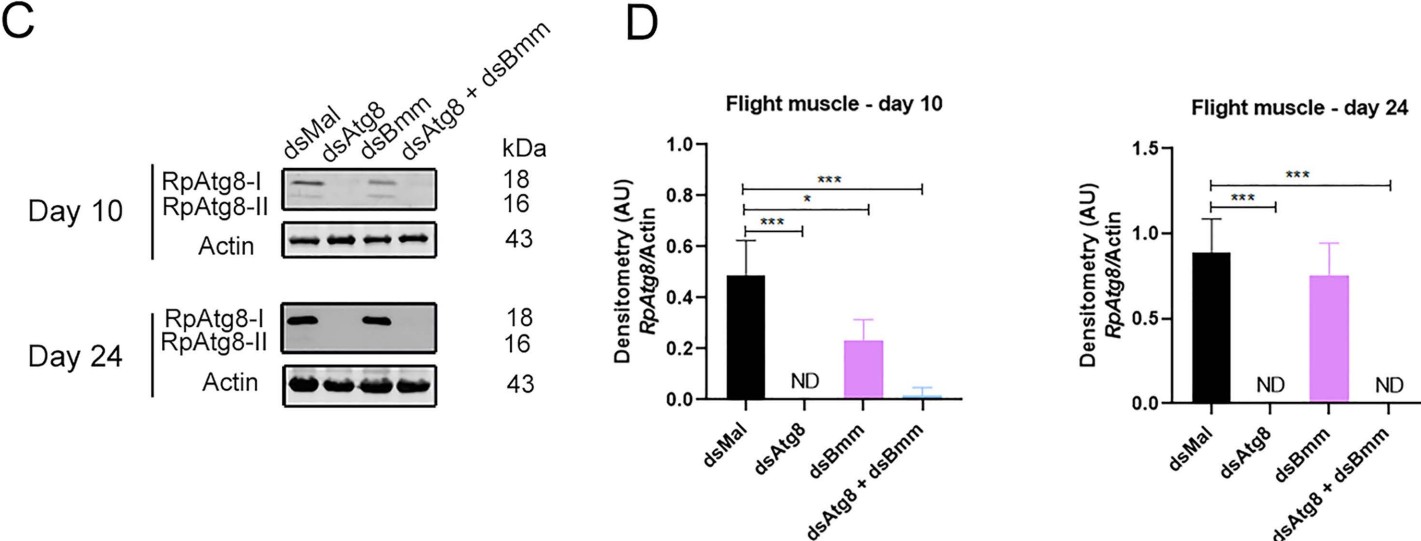

**Fig 4. Silencing efficiencies of *RpAtg8* and *RpBmm* in the flight muscle.** Adult females, 18 days after a blood meal, were injected with 1 μg of dsRNA for *RpAtg8, RpBmm, RpAtg8+RpBmm*, or *Mal* (control), fed three days later, and dissected either five or ten days after feeding (Protocol 1) or injected on the tenth day after feeding and dissected 14 days after injection (Protocol 2) (n=4-7). **(A–B)** mRNA levels were determined by qPCR, using *Rp18S* expression as a reference gene. *RpAtg8* and *RpBmm* mRNA quantification in the fat body. Gene expression levels are relative to each control value (dashed lines). **(C)** RpAtg8 immunoblotting in the flight muscle on the 10th day (top image) and the 24th day (bottom image) (n=3). **(D)** Total RpAtg8 densitometry on the 10th and 24th days. RpAtg8-I: free RpAtg8. RpAtg8-II: lipidated RpAtg8. All graphs show mean±SEM. *p<0.05, **p<0.01, ***p<0.001, ****p<0.0001, when compared by one-way ANOVA followed by Tukey's post-test.

# Flight Muscle

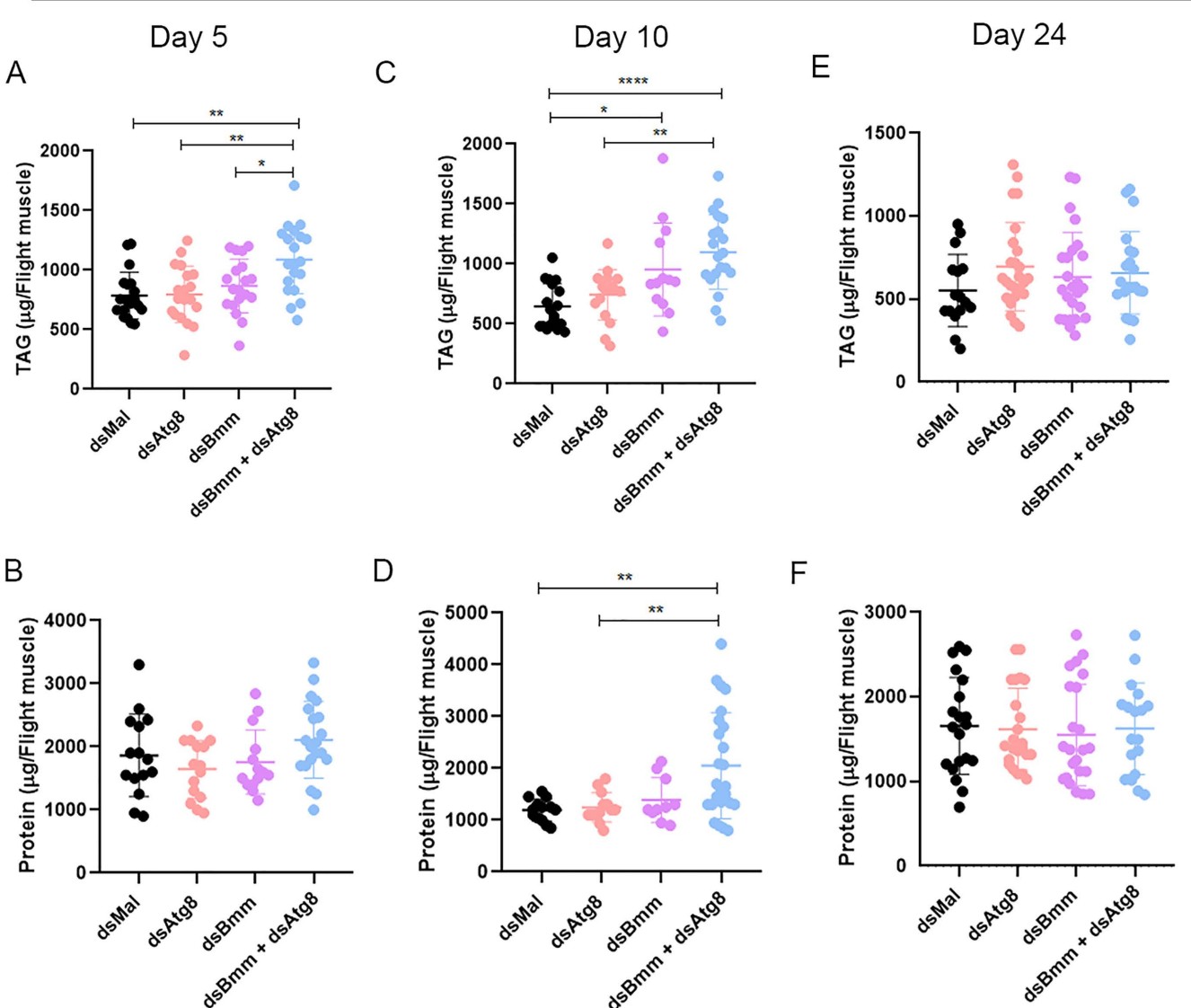

**Fig 5. Silenced females accumulate TAG in the flight muscle during starvation.** Adult females (18 days after a blood meal) were injected with 1 µg of dsRNA for *RpAtg8, RpBmm, RpAtg8+RpBmm*, or *Mal* (control), fed three days later, and dissected either five or ten days after feeding (Protocol 1) or injected on the tenth day after feeding and dissected 14 days after injection (Protocol 2). Flight muscles were collected, washed, individually homogenized, and total TAG **(A, C, E)** and protein content **(B, D, F)** on the 5th (A, B), 10th (C, D), and 24th (E, F) day after the blood meal were determined. Graphs show mean ± SD (n = 12–23 insects, obtained from 3 independent experiments). *p < 0.05, **p < 0.01, ****p < 0.0001, when compared by one-way ANOVA followed by Tukey's post-test.

20% and 50% reduced expression of *AKHr* and *Dgat2* respectively at day 5, and 1.5-fold and 2.5-fold upregulations of *RpAtg8* and *Dgat1*, respectively, on day 10 (Fig 6B and S3 Fig). On day 24, high variations were observed, but tendencies in the upregulation of Dgat1 and *Plin1* were observed. Under dual silencing conditions, *AKHr* and *Dgat2* were significantly downregulated on day 5, and *Acc* expression was reduced on day 24 (Fig 6C and S3 Fig).

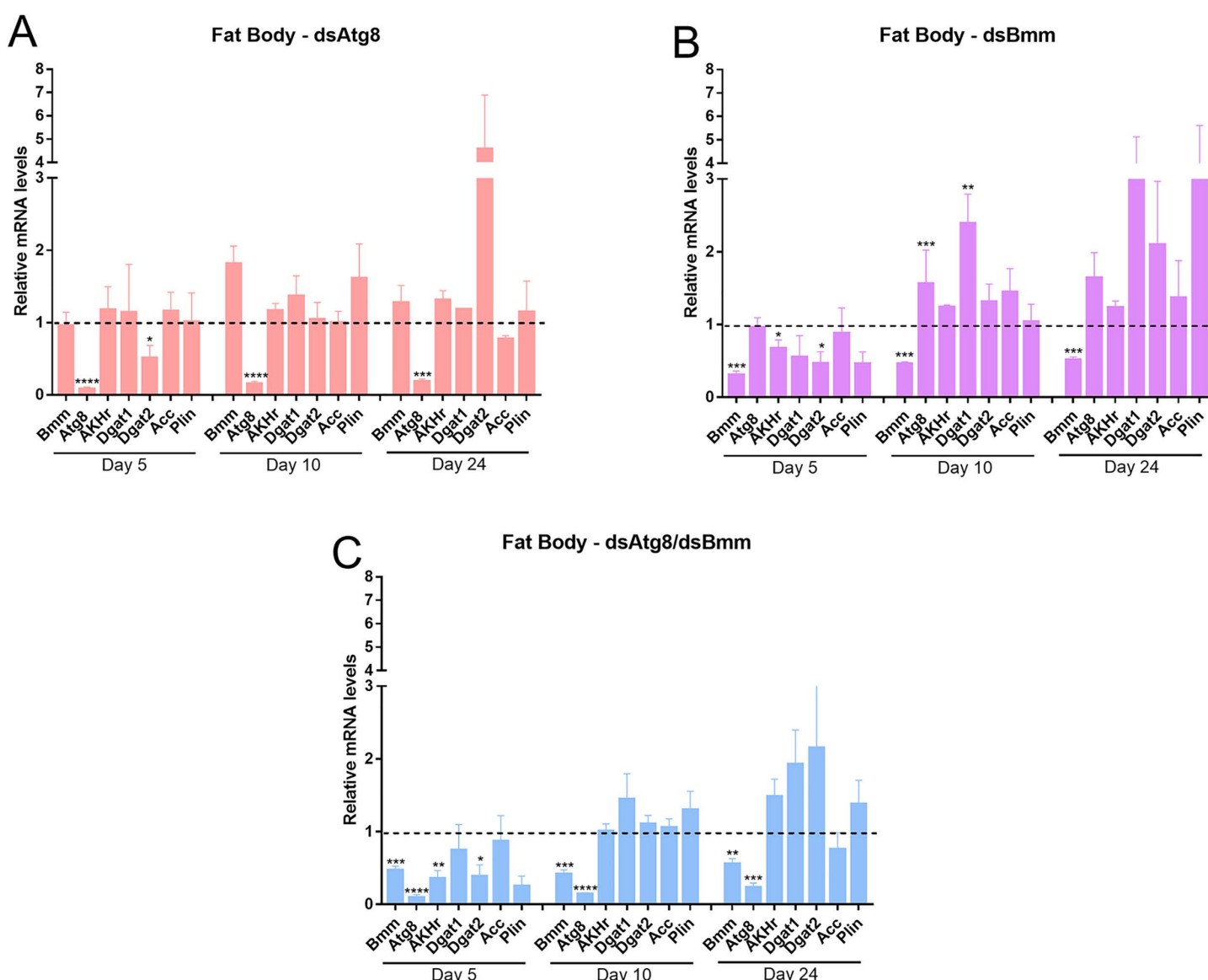

**Fig 6. Silencing of *RpAtg8* and *RpBmm* affects the expression of lipid metabolism-related genes in the fat body.** Adult females (18 days after a blood meal) were injected with 1 μg of dsRNA for *RpAtg8, RpBmm, RpAtg8+RpBmm*, or *Mal* (control), fed three days later, and dissected either five or ten days after feeding (first protocol) or injected on the tenth day after feeding and dissected 14 days after injection (second protocol). Gene expression levels in the fat body were determined by qPCR using specific primers designed to target different genes. *Rp18S* amplification was used as an endogenous control. Gene expression levels are relative to each control value (dashed line). The graphs show mean±SEM of 5 independent determinations, n=5. *p<0.05, **p<0.01, ***p<0.001, ****p<0.0001, when compared by Student's t-test. Akhr, adipokinetic hormone receptor; ACC, acetyl-CoA carboxylase; DGAT1, diacylglycerol acyltransferase 1; DGAT2, diacylglycerol acyltransferase 2; Plin, perilipin.

In the flight muscle, silencing of *RpAtg8* alone led to a dynamic modulation of Plin1 expression, with upregulation observed at day 5, followed by significant downregulation on days 10 and 24 (Fig 7A). *Dgat1* and *Dgat2* exhibited an upward trend in expression on days 5 and 24, although these changes were not statistically significant. Silencing of *RpBmm* resulted in decreased expression of both *AKHr* and *Plin1* on day 5, with continued downregulation of *Plin1* on day 10 (Fig 7B). Under dual silencing conditions, only *Acc* and *Plin1* showed significant downregulation, specifically on day 10 (Fig 7C).

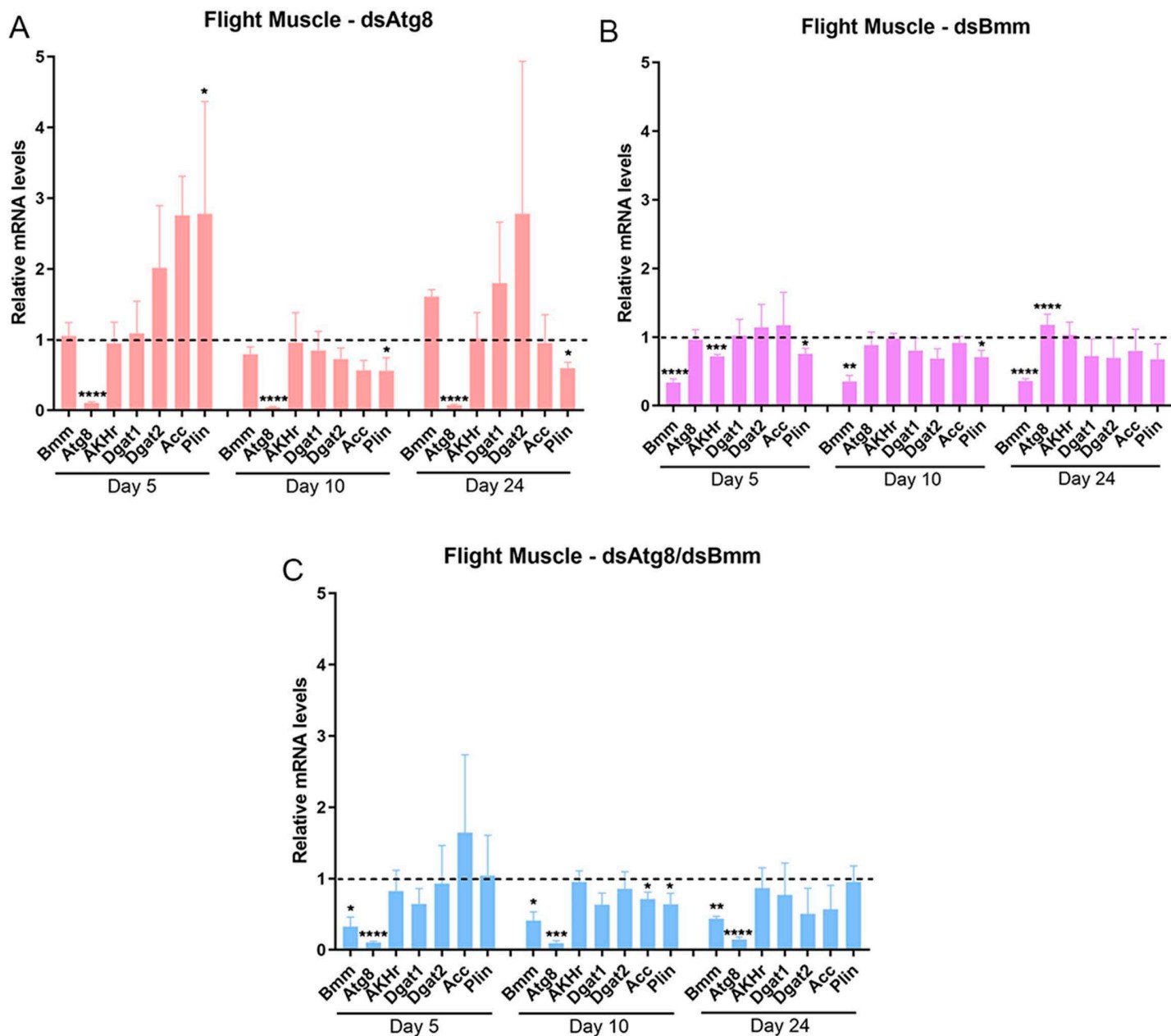

**Fig 7. Silencing of *RpAtg8* and *RpBmm* affects the expression of lipid metabolism-related genes in the flight muscle.** Adult females (18 days after a blood meal) were injected with 1 μg of dsRNA for *RpAtg8, RpBmm, RpAtg8+RpBmm*, or *Mal* (control), fed three days later, and dissected either five or ten days after feeding (first protocol) or injected on the tenth day after feeding and dissected 14 days after injection (second protocol). Gene expression levels in the flight muscle were determined by qPCR using specific primers designed to target different genes. *Rp18S* amplification was used as an endogenous control. Gene expression levels are relative to each control value (dashed line). The graphs show mean ± SEM of 5 independent determinations, n = 5. *p < 0.05, **p < 0.01, ***p < 0.001, ****p < 0.0001, when compared by Student's t-test. Akhr, adipokinetic hormone receptor; ACC, acetyl-CoA carboxylase; DGAT1, diacylglycerol acyltransferase 1; DGAT2, diacylglycerol acyltransferase 2; Plin, perilipin.

Collectively, these findings suggest that disruption of lipophagy and lipolysis pathways induces distinct and temporally regulated transcriptional responses in genes associated with lipid metabolism, in a tissue-specific manner.

### 3.4. Simultaneous silencing of *RpAtg8* and *RpBmm* impairs reproduction and survival, but without additive effects

To ensure that any observed phenotypes were not secondary to impaired digestion, we first evaluated the digestion of silenced insects. No significant differences in midgut protein contents, which reflects overall digestion, were found among any of the silenced groups when compared to controls along the first 10 days after the blood meal (Fig 8A). The fat body is a key tissue for vitellogenin (Vg) synthesis, which is secreted into the hemolymph and subsequently taken up by developing oocytes (Engelmann, 1979). Notably, total protein content in the hemolymph presented a trend toward reduction in all silenced groups, *RpAtg8*, *RpBmm*, and the double-silenced insects, when compared to control on day 5 after the blood meal (Fig 8B). Given its pivotal role in reproduction, Vg levels in the hemolymph were analyzed by SDS-PAGE. The fragments representing Vg subunits have been previously characterized [61]. Both the representative gel (Fig 8C, arrowheads point to Vg subunits) and the densitometric analysis (Fig 8D) revealed a significant reduction in Vg levels in all silenced groups on day 5 after the blood meal. Considering the importance of lipid and protein reserves for successful oogenesis and oviposition, we next examined reproductive outputs. Silencing efficiencies were tested and showed at least 60% reductions in the ovaries (S4A, S4B Fig). Oviposition was significantly reduced by 20–30% in all silenced groups (Fig 8E), and egg/embryo viability was compromised, as evidenced by a marked reduction in hatching rates of approximately 70% in the *Bmm* alone and double silenced groups (Fig 8F). As previously demonstrated, the silencing of *RpAtg8* alone triggers modest reductions in the hatching rates [49]. Interestingly, *RpAtg8*-silenced females showed a moderate but sustained decline in hatching, while eggs from *RpBmm*- and double-silenced females displayed lower hatching rates from day 6 onwards (S4C Fig). Lastly, adult longevity was also affected, with all silenced groups exhibiting a significant reduction in survival rates (Fig 8G). These findings underscore the essential role of lipolysis and lipophagy in maintaining reproductive fitness and survival in *R. prolixus*. However, the absence of additive effects in the double-silenced group suggests that these pathways do not compensate for each other to maintain lipid homeostasis under the tested conditions.

### 3.5. Dual silencing of *RpAtg8* and *RpBmm* leads to the accumulation of larger lipid droplets in the fat body

Although TAG levels in the double knockdown group were not different from those observed in the single knockdowns (Fig 3), we still investigated whether lipid droplet (LD) size might be affected. To address this, we analyzed Nile Red-stained fat body images collected at three time points across the four experimental groups. Representative images (Fig 9) showed clear alterations in LD morphology in knockdown insects relative to controls. To quantify these changes, we measured the maximum LD diameters and generated frequency distribution histograms. On days 5, 10, and 24 after the blood meal, all silenced groups exhibited significantly enlarged LDs compared to controls (Fig 10A–10C). Notably, on days 5 and 24, the dual knockdown triggered a more pronounced increase in LD size than either of the single knockdowns (Fig 10A and 10C). This shift toward larger LDs was also evident in the distribution histograms. Interestingly, on day 10, LD enlargement in the dual knockdown group was less pronounced than in the single knockdowns (Fig 10B), suggesting dynamic changes in LD remodeling.

These findings indicate that, although no further TAG accumulation was observed in the fat body of double knockdown insects, LD morphology is still modulated, potentially as an adaptive mechanism to maintain lipid homeostasis under these conditions. Moreover, the distinct patterns observed across time points suggest that the regulation of lipid degradation pathways, LD remodeling, and overall lipid metabolism varies with the nutritional status of the insect.

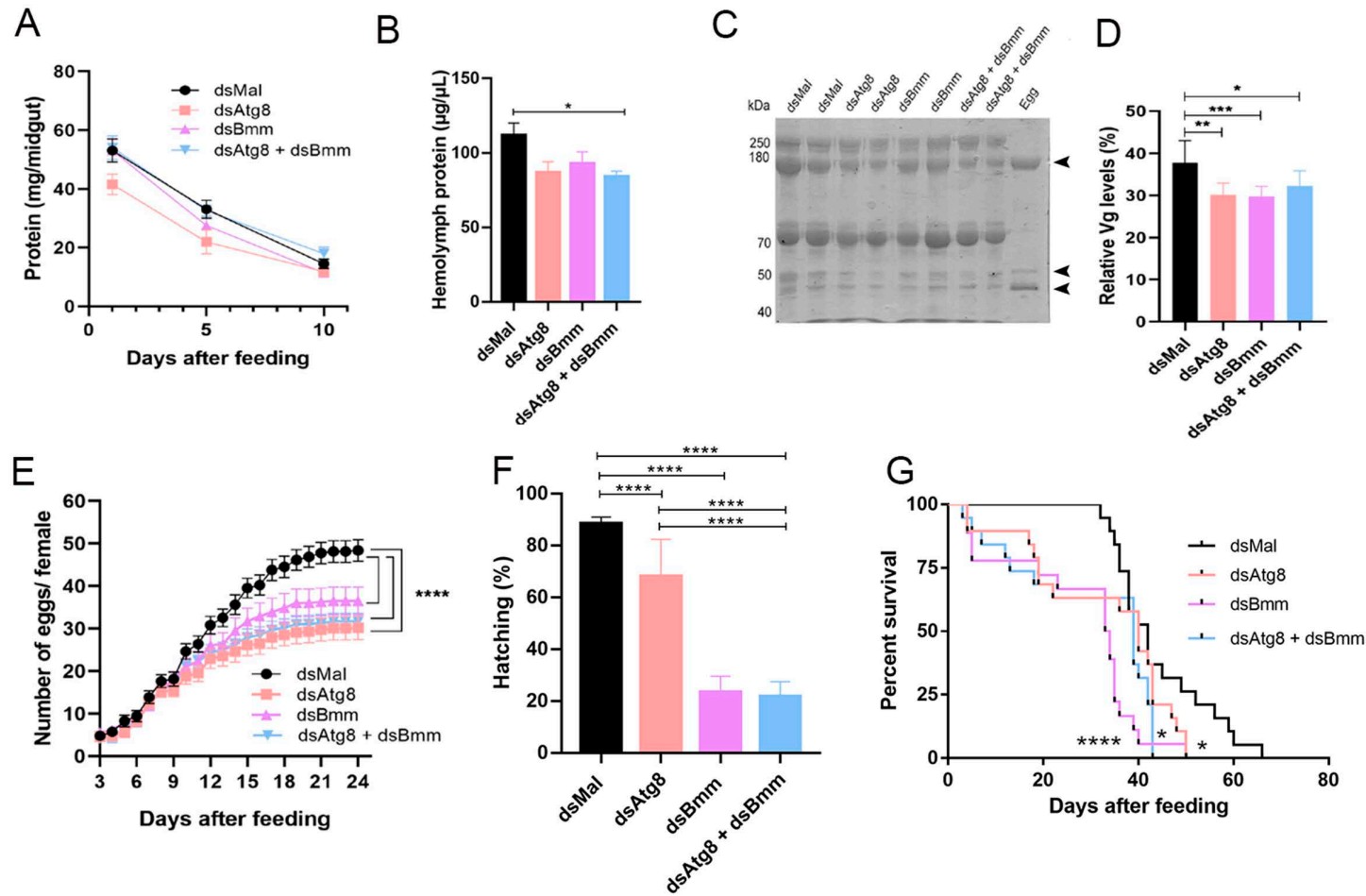

**Fig 8. Silencing of *RpAtg8* and *RpBmm* affects physiological aspects of silenced insects.** Adult females (18 days after a blood meal) were injected with 1 μg of dsRNA for *RpAtg8, RpBmm, RpAtg8+RpBmm*, or *Mal* (control), fed three days later, and dissected on different days after feeding. **(A)** On different days after the blood meal, insects were dissected and total protein content in the midgut was determined. Symbols represent mean±SD (n=10–13). **(B)** Total protein concentration in the hemolymph was measured on day 5 after feeding. Results are shown as mean±SD (n=5–7), compared by one-way ANOVA followed by Tukey's post-test **(C–D)** Protein profile of the hemolymph (0.7 μL) by SDS-PAGE (7.5% polyacrylamide) followed by Vg protein densitometry on day 5 after feeding. An egg sample was used to confirm the identification of Vg bands. Arrowheads indicate Vg subunits. Results are shown as mean±SD (n=8), compared by one-way ANOVA followed by Tukey's post-test. **(E)** Oviposition rates monitored daily after the blood meal. Results show mean±SD (n=17). (****): p<0.0001 analyzed by two-way ANOVA followed by Sidak's post-test. **(F)** Total hatching rates were determined, n=855. (****): p<0.0001 analyzed by the χ2 test. **(G)** Survival was monitored daily (n=18-19, in three independent experiments), being (*) and (****): p<0.05 and p<0.0001, respectively, after analysis by the Log-rank test (Mantel-Cox).

### 3.6. Combined silencing of *RpAtg8* and *RpBmm* does not lead to additive effects in forced flight capacity

To further investigate whether the simultaneous reduction of RpAtg8 and RpBmm affects insect physiology, we subjected silenced insects to high-performance activity using a forced flight assay. In this experiment, insects were individually suspended by a thread in front of a tube delivering continuous airflow and forced to fly until exhaustion (Oliveira et al., 2006). Under fed conditions (day 10 after the blood meal), silencing either *RpAtg8* or *RpBmm* individually led to a marked reduction in flight capacity, with average flight times dropping from approximately 40 minutes in controls to less than 20 minutes in the knockdown groups. Under starvation (day 24 after the blood meal), a physiological state in which *R. prolixus* is more prone to flight [59], a more significant decrease in flight capacity was observed in all knockdown groups, consistent with

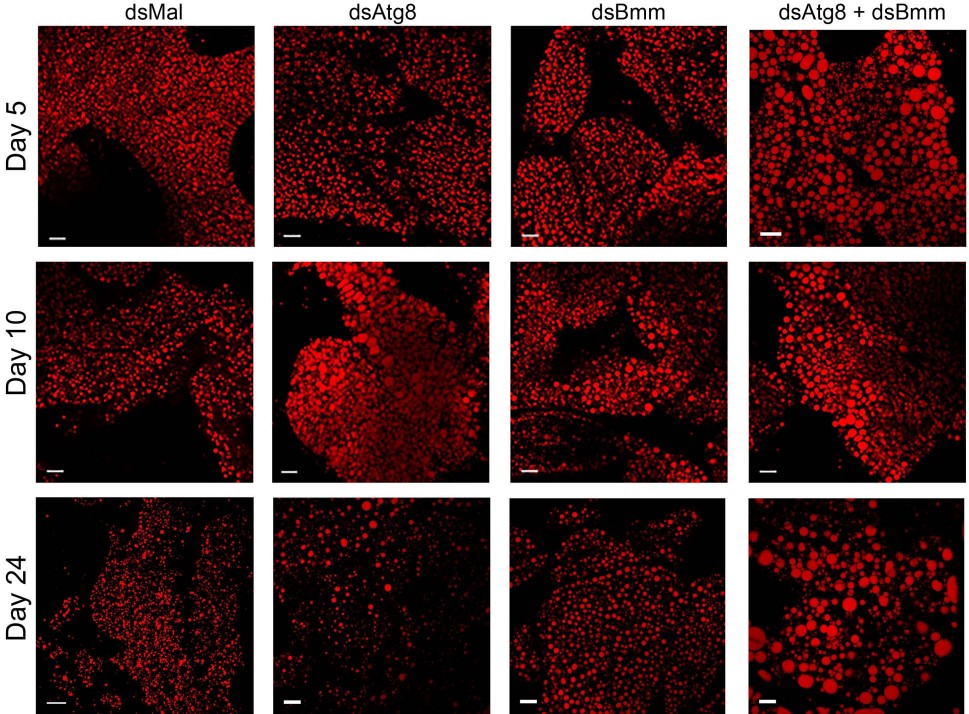

**Fig 9. Dual silencing of *RpAtg8* and *RpBmm* leads to additive alterations in LDs diameters in the fat body.** Adult females (18 days after a blood meal) were injected with 1 µg of dsRNA for *RpAtg8, RpBmm, RpAtg8+RpBmm*, or *Mal* (control), fed three days later, and dissected either five or ten days after feeding (Protocol 1) or injected on the tenth day after feeding and dissected 14 days after injection (Protocol 2). Lipid droplets (LDs) from freshly dissected fat bodies on days 5, 10, and 24 after the blood meal were stained with Nile Red and observed under a confocal laser scanning microscope. DAPI-stained nuclei were also observed. Images are representative from three independent experiments (n = 3). Bars: 40 µm.

previous reports [27,47]. While control insects flew for an average of 50 minutes, knockdown insects exhibited flight durations below 20 minutes (Fig 11B). However, for both conditions, the double knockdown did not result in a further reduction in flight capacity when compared to the individual knockdowns (Fig 11A, 11B).

## 4. Discussion

*Rhodnius prolixus* is a hematophagous insect and a primary vector of *Trypanosoma cruzi*, the etiological agent of Chagas disease. Its unique physiology, particularly its reliance on blood-derived nutrients and long fasting periods between feedings, makes it a valuable model for studying energy homeostasis and lipid metabolism [8,62]. In this work, we performed the simultaneous silencing of two key genes involved in lipid mobilization: *RpAtg8*, required for autophagy (lipophagy), and *RpBmm*, which encodes a brummer lipase ortholog, required for lipolysis. While individual knockdowns of these genes led to well-documented changes in lipid mobilization and reproductive capacity [27,47,48], we found that the dual knockdown did not result in additive effects on most parameters, particularly TAG accumulation in the fat body, which was our primary readout of lipid storage and mobilization. This outcome challenged our initial hypothesis that lipolysis and lipophagy were independent pathways that could act in a compensatory manner to sustain energy balance. In the following sections, we will discuss our thoughts when trying to interpret our observations, in accordance with data from the literature.

While classical studies in mammals have established that lipophagy contributes to lipid degradation alongside neutral lipolysis [36,37,63], investigations exploring the effects of simultaneous inhibition of both pathways remain scarce. To our

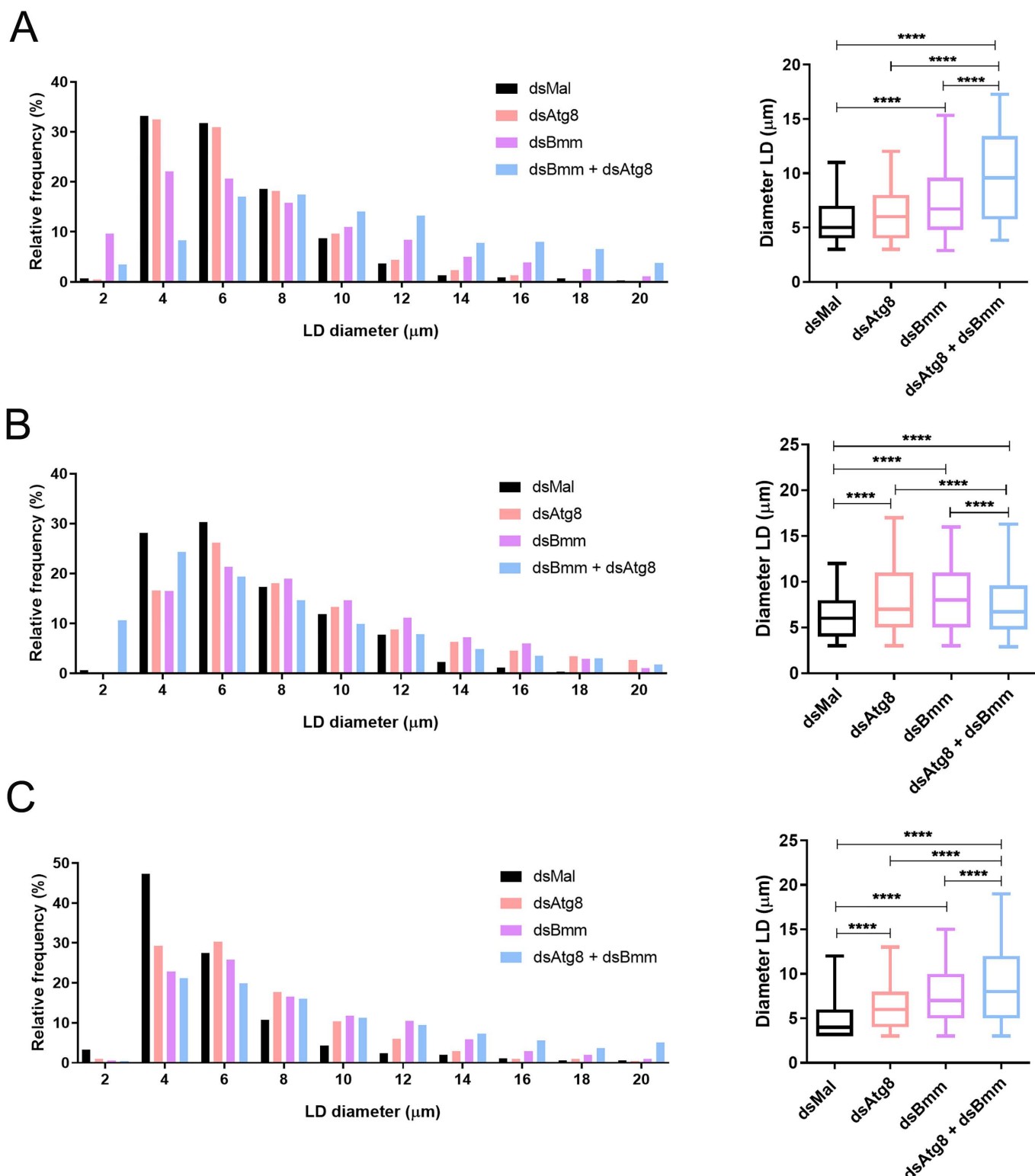

**Fig 10. Quantifications of the additive alterations in LDs diameters in the fat body.** Adult females (18 days after a blood meal) were injected with 1 µg of dsRNA for *RpAtg8, RpBmm, RpAtg8+RpBmm*, or *Mal* (control), fed three days later, and dissected five or ten days after feeding (Protocol 1), or injected on day ten after feeding and dissected 14 days after injection (Protocol 2). **(A)** Histogram of LD diameter distribution and quantification of

maximum LD diameters on day 5. **(B)** Histogram of LD diameter distribution and quantification of maximum LD diameters on day 10. **(C)** Histogram of LD diameter distribution and quantification of maximum LD diameters on day 24. Three experiments were performed using insects from different cohorts (n = 3), and 2 images from each experiment were quantified. The graphs show the medians ± 5th-95th percentiles of at least 2000 LDs per condition. ****p < 0.0001, when compared by one-way ANOVA followed by Tukey's post-test.

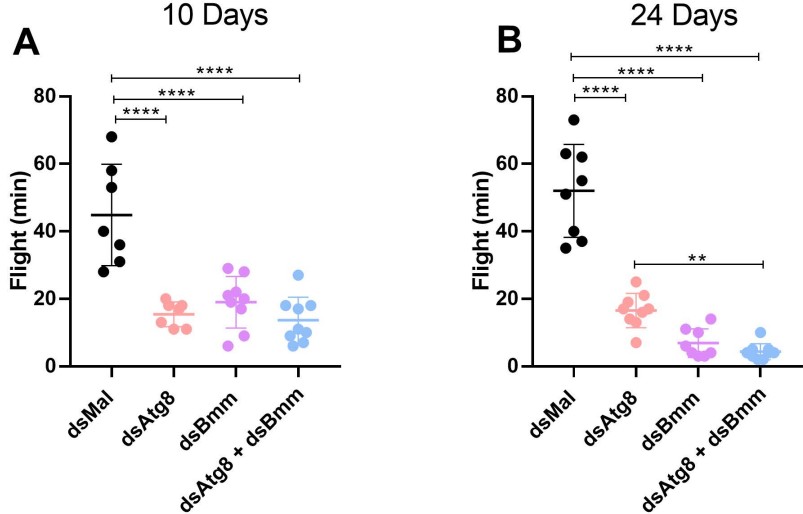

**Fig 11. Dual silencing of *RpAtg8* and *RpBmm* leads to reduced flight capacity.** Adult females (18 days after a blood meal) were injected with 1 µg of dsRNA for *RpAtg8, RpBmm, RpAtg8 + RpBmm*, or *Mal* (control), fed three days later (Protocol 1), or injected on day ten after feeding (Protocol 2). **(A)** Following protocol 1, insects were subjected to a forced flight test on day 10 after feeding. The duration of flight activity until exhaustion was recorded individually (n = 7-9). **(B)** Following protocol 2, insects were subjected to a forced flight test on day 24 after feeding. The duration of flight activity until exhaustion was recorded individually (n = 8-9). Graphs show mean ± SDs. **p < 0.01 and ****p < 0.0001 when compared by one-way ANOVA followed by Tukey's post hoc test.

knowledge, to date only two studies have directly addressed this issue. In rat hepatocytes, dual inhibition of ATGL and lysosomal acid lipase (LAL) produced a phenotype resembling ATGL inhibition alone, marked by the accumulation of large lipid droplets (LDs). These results suggest that lipolysis acts upstream of lipophagy, targeting specific LD pools first [64]. In contrast, in zebrafish, knockout models for ATGL and Atg8 both exhibited impaired growth, reduced locomotor activity, and lower oxygen consumption. ATGL-deficient fish accumulated TAG in the liver and formed enlarged hepatic LDs without upregulating lipophagy, whereas Atg8-deficient fish showed reduced total lipid storage, increased hepatic cholesterol, and upregulation of compensatory lipolysis, particularly *Atgl* expression. When both ATGL and autophagy were inhibited simultaneously, an additive accumulation of TAG was observed in the liver [65]. These findings reinforce the notion that, although the outcomes vary by model, lipolysis and lipophagy are coordinated in tissue- and context-specific ways.

Our findings in *R. prolixus* add to this emerging picture by revealing that simultaneous knockdown of *RpAtg8* and *RpBmm* does not result in an additive phenotype regarding TAG accumulation. In fact, similar levels of TAG were observed when either pathway was silenced alone or in combination. To interpret this non-additive phenotype, it is essential to consider how these two pathways may operate. The possibilities include: (1) hierarchical organization, in which one pathway acts upstream of the other; (2) targeting distinct LD subpopulations by each pathway; or (3) redundancy or compensatory activity between the two.

If lipolysis and lipophagy act hierarchically and target specific LD pools, the phenotype of the dual knockdown would resemble that of the upstream pathway. This is consistent with the findings in rat hepatocytes described above [64], where

inhibition of lipolysis alone mimicked the dual inhibition phenotype, likely due to disruption of the upstream pathway. Moreover, if each pathway targets LDs with specific characteristics, such as size, composition, or associated proteins, the phenotypic outcome would depend on which subset remains functional. In mammalian models, differences in LD size have served as a proxy to identify the preferential targeting by each pathway. However, in *R. prolixus*, we did not observe clear size-based targeting preferences between the pathways. It is possible that the distinguishing features of LD subpopulations in this case are not related to size and may not be discernible using conventional Nile Red staining.

Conversely, if the pathways were functionally independent and complementary, we would expect an additive phenotype upon dual inhibition, which was not the case. This supports a model of functional coordination between lipophagy and lipolysis in *R. prolixus*, possibly through hierarchical regulation. It is important to note, however, that our interpretations are limited by the partial silencing efficiency of RNAi. Residual activity of either *RpAtg8* or *RpBmm* could be sufficient to maintain basal TAG turnover, masking more severe phenotypes. Future studies using stable knockouts in genetically tractable models such as *Drosophila melanogaster*, or employing CRISPR-based approaches in *R. prolixus*, will be essential to further test the consequences of complete inhibition.

Interestingly, although TAG levels in the fat body were not further increased in the double knockdown group, LDs were significantly enlarged compared to the single knockdowns, indicating a compensatory remodeling response. LDs are dynamic organelles that vary in number, size, and distribution, and can grow through fusion, lipid synthesis, or lipid transfer. All these processes are influenced by metabolic signals and by LD composition [66,67]. To our knowledge, direct evidence of LD fusion in insects is not available. However, studies in *Drosophila* S2 cells suggest that phospholipid metabolism influences LD size regulation [68,69], and that phosphatidylcholine availability and phospholipid remodeling can drive LD fusion [70].

As mentioned before, in mammals, LD size is linked to mobilization efficiency, with larger LDs favoring lipolysis and smaller ones more accessible to autophagy [64,71]. It is possible that LD remodeling, whether in size or other features, acts as a signal directing which degradation pathway is engaged. Although such mechanisms remain less explored in insects, our findings suggest that similar principles may regulate LD utilization in *R. prolixus* and warrant further investigation.

Adding to the complexity, we observed modulation in the expression of lipid metabolism-related enzymes across tissues and nutritional conditions. Although transcriptional data do not confirm changes in enzymatic activity, they suggest that compensatory mechanisms in lipid metabolism could contribute to the observed phenotypes, possibly buffering the effects of the dual knockdown.

Beyond lipid-related pathways, it is also possible that metabolic compensation occurs through alternative routes. Given the protein-rich nature of the blood meal, *R. prolixus* may shift toward amino acid catabolism and/or gluconeogenesis in response to impaired lipid mobilization. Such plasticity could preserve systemic energy balance and obscure phenotypes related to TAG accumulation. This underscores the metabolic flexibility of hematophagous insects and highlights the challenges of interpreting phenotypes in such a complex system.

In the flight muscle, previous studies have shown abundant LDs and high TAG content in *R. prolixus* [72,73], with lipids serving as a major energy source during flight, delivered via the lipophorin shuttle or derived from local TAG stores [8,59]. Functional studies have demonstrated that silencing *RpBmm* or *RpAtg8* impairs flight performance [7,27,47,48]. Curiously, in our current study, although dual knockdown led to additional TAG accumulation in the flight muscle by day 5 post-feeding, no difference in forced flight capacity was observed. This suggests that the mobilization of locally stored TAG is not the limiting factor for flight performance in fed insects, and other factors may compensate to maintain this function.

In summary, our study demonstrates that lipid metabolism in *R. prolixus* is remarkably plastic and regulated by context-specific and possibly hierarchical mechanisms. Despite the absence of additive effects on TAG accumulation following the simultaneous silencing of *RpAtg8* and *RpBmm*, our results show significant alterations in LD architecture. These findings reveal that lipophagy and lipolysis are likely engaged through distinct regulatory cues and contribute to the

complexity of lipid homeostasis in insects. Advancing our understanding of these metabolic interactions not only enhances fundamental insect physiology but may also identify metabolic vulnerabilities exploitable for vector control strategies.

## Supporting information

**S1 Fig. Representative segmented images of lipid droplets used for quantification.** Examples of DAIME-based segmentation are shown for (A) day 10 dsAtg8 and (B–C) two additional representative experimental conditions. Lipid droplets were stained with Nile Red and segmented using the Marr-Hildreth edge detection algorithm with watershed set to 12% and exclusion of dark regions. No manual refinement was required. These images illustrate the basis for the lipid droplet diameter quantification presented in the manuscript.
(TIF)

**S2 Fig. Full, uncropped immunoblots corresponding to Figs 2C and 4C.** Uncropped images of the original blots are shown to document the full experimental results. The specific regions used in the main figures are indicated in red.
(TIF)

**S3 Fig. (Graphs with non-segmented y-axes) Silencing of *RpAtg8* and *RpBmm* affects the expression of lipid metabolism-related genes in the fat body.** Adult females (18 days after a blood meal) were injected with 1 µg of dsRNA for *RpAtg8, RpBmm, RpAtg8 + RpBmm*, or *Mal* (control), fed three days later, and dissected either five or ten days after feeding (first protocol) or injected on the tenth day after feeding and dissected 14 days after injection (second protocol). Gene expression levels in the fat body were determined by qPCR using specific primers designed to target different genes. *Rp18S* amplification was used as an endogenous control. Gene expression levels are relative to each control value (dashed line). The graphs show mean ± SEM of 5 independent determinations, n = 5. *p < 0.05, **p < 0.01, ***p < 0.001, ****p < 0.0001, when compared by Student's t-test. Akhr, adipokinetic hormone receptor; ACC, acetyl-CoA carboxylase; DGAT1, diacylglycerol acyltransferase 1; DGAT2, diacylglycerol acyltransferase 2; Plin, perilipin.
(TIF)

**S4 Fig. Silencing of *RpAtg8* and *RpBmm* in the ovary on days 5 and 10 after the blood meal.** Adult females (18 days after a blood meal) were injected with 1 µg of dsRNA for *RpAtg8, RpBmm, RpAtg8 + RpBmm*, or *Mal* (control), fed three days later, and dissected five or ten days after feeding (protocol 1). **(A–B)** Quantification of *RpAtg8* and *RpBmm* mRNA in the ovary. mRNA levels were determined by qPCR, using *Rp18S* expression as a reference gene. **(C)** Hatching proportions of eggs laid on different days after feeding. Graphs show mean ± SEM (n = 4). *p < 0.05, **p < 0.01, ***p < 0.001, ****p < 0.0001, when compared by one-way ANOVA followed by Tukey's post-test.
(TIF)

**S1 Table. Genes and primers list.** Sequences of primers used in qPCR experiments and dsRNA synthesis. Sequences were obtained from VectorBase (https://www.vectorbase.org/) or previous works and synthesized by Macrogen or IDT technologies.
(DOCX)

## Acknowledgments

The authors thank Bruna Beatris Santana Afonso, Maysa Moura Lopes and Geane Cleia Pereira Braz for insect handling and care. Image acquisition using the LEICA SPE was perfomed at the Unidade de Microscopia ICB/UFRJ.

## Author contributions

**Conceptualization:** Samara Santos-Araujo, Katia C. Gondim, Isabela Ramos.

**Data curation:** Samara Santos-Araujo.

**Formal analysis:** Samara Santos-Araujo, Katia C. Gondim, Isabela Ramos.

**Funding acquisition:** Katia C. Gondim, Isabela Ramos.

**Supervision:** Katia C. Gondim, Isabela Ramos.

**Writing – original draft:** Katia C. Gondim, Isabela Ramos.

**Writing – review & editing:** Samara Santos-Araujo, Katia C. Gondim, Isabela Ramos.

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
