## [Decision Letter · Decision Letter 0]

17 Sep 2025

Dear Dr. Ramos,

Thank you for submitting your manuscript to PLOS ONE. After careful consideration, we feel that it has merit but does not fully meet PLOS ONE’s publication criteria as it currently stands. Therefore, we invite you to submit a revised version of the manuscript that addresses the minor points raised during the review process.

We look forward to receiving your revised manuscript.

Kind regards,

Monika Oberer

Academic Editor

PLOS ONE

**Journal Requirements:**

1. When submitting your revision, we need you to address these additional requirements. Please ensure that your manuscript meets PLOS ONE's style requirements, including those for file naming. The PLOS ONE style templates can be found at https://journals.plos.org/plosone/s/file?id=wjVg/PLOSOne_formatting_sample_main_body.pdf and https://journals.plos.org/plosone/s/file?id=ba62/PLOSOne_formatting_sample_title_authors_affiliations.pdf 2. Please update your submission to use the PLOS LaTeX template. The template and more information on our requirements for LaTeX submissions can be found at http://journals.plos.org/plosone/s/latex. 3. PLOS ONE now requires that authors provide the original uncropped and unadjusted images underlying all blot or gel results reported in a submission’s figures or Supporting Information files. This policy and the journal’s other requirements for blot/gel reporting and figure preparation are described in detail at https://journals.plos.org/plosone/s/figures#loc-blot-and-gel-reporting-requirements and https://journals.plos.org/plosone/s/figures#loc-preparing-figures-from-image-files. When you submit your revised manuscript, please ensure that your figures adhere fully to these guidelines and provide the original underlying images for all blot or gel data reported in your submission. See the following link for instructions on providing the original image data: https://journals.plos.org/plosone/s/figures#loc-original-images-for-blots-and-gels.   In your cover letter, please note whether your blot/gel image data are in Supporting Information or posted at a public data repository, provide the repository URL if relevant, and provide specific details as to which raw blot/gel images, if any, are not available. Email us at plosone@plos.org if you have any questions. 4. Please include captions for your Supporting Information files at the end of your manuscript, and update any in-text citations to match accordingly. Please see our Supporting Information guidelines for more information: http://journals.plos.org/plosone/s/supporting-information. 5. If the reviewer comments include a recommendation to cite specific previously published works, please review and evaluate these publications to determine whether they are relevant and should be cited. There is no requirement to cite these works unless the editor has indicated otherwise. 

Reviewers' comments:

**Comments to the Author**

1. Is the manuscript technically sound, and do the data support the conclusions?

Reviewer #1: Yes

Reviewer #2: Yes

2. Has the statistical analysis been performed appropriately and rigorously?

Reviewer #1: Yes

Reviewer #2: Yes

3. Have the authors made all data underlying the findings in their manuscript fully available?

Reviewer #1: Yes

Reviewer #2: Yes

4. Is the manuscript presented in an intelligible fashion and written in standard English?

Reviewer #1: Yes

Reviewer #2: Yes

**Reviewer #1: ** In this manuscript, Santos-Araujo et al. investigate how lipolysis and lipophagy regulate lipid homeostasis in the Chagas disease related insect Rhodnius prolixus. Using RNAi technology, they downregulated RpBmm (a Brummer/ATGL ortholog) and RpAtg8 (autophagy-related) under fed and starved conditions. They measured TAG in fat body and flight muscle, studied LD morphology and well as expression of lipid genes and provided some fitness readouts. One major finding is that dual knockdown does not further increase fat-body TAG compared to single knockdowns but is accompanied by morphological alteration of LDs. These results support the notion of a tissue and state dependent coordination between lipolysis and lipophagic. They highlight the metabolic plasticity of these processes. The study is of broader interest; however, certain aspects of the TAG assay normalization and LD size quantification raise concerns.

Concerns,

(1) The TAG content of the fat bodies was analysed using a Kit and per fat body.

* Why was the TAG content not normalized to fat-body wet mass to control for preparation-related variability (e.g., tissue recovery), in line with previous recommendations (Tennessen et al., Methods 68, 2014, 105–115)? Please clarify.

* In this context, where insect size and/or fat body size different between groups?

(2) The authors used Nile Red labeling for image-based quantification of lipid droplet diameters. They used images of “peripheral regions” of fat bodies. Apparently, the imaged lobes can show different LD size/content (e.g. Figure 3 day 24 dsRNA). Furthermore, in the single images, it is likely that some LDs were insufficiently acquired (those partly out of focus), which could additionally limit the analysis.

* What are the criteria for the selection of the areas?

* How much of the prepared fat body area do the images approximately represent (10%, 50 %....)?

(3) The authors measured the diameter of LDs to highlight phenotypic differences across the different conditions. The authors cited the paper of Michele Alves-Bezerra et al. as a reference for the LD quantification process using the DAIME software, which is software to detect cells of phytoplankton, here used for LD quantification.

This paper, however, only links to the original paper (H. Daims, et al. daime, a novel image analysis program for microbial ecology and biofilm research, Environmental Microbiology 8 (2006) 200–213). In turn, in the original paper it is stated that the “edge detection” algorithm originates from the method introduced by Viles and Sieracki, 1992.

The latter evaluated different algorithms for marine plankton cell detection, not for lipid droplets. The best algorithm for the specific application was the Marr-Hildreth method with “edge strength” for automated object edge detection. Likely, this method is similar to a Laplacian of Gaussian edge detection with additional enhancement steps. Since such methods detect more abrupt contrast changes of image objects, it can be very challenging to detect and separate very closely associated LDs and putatively imaged with decreased optical resolution (20x objective).

It is not clear that this method can separate LDs in the provided images sufficiently for quantification (assuming that the overall rules of image acquisition were considered, e.g. no significant overexposed pixels, etc.).

* Please show as supplementary data the segmented image of the representative image of day 10 dsATG8 and of two other examples. These images are the basis of the quantification outcome. One can export the segmented data by right-clicking on the image thumbnail and selecting “extract object layer.” If manual refinements were necessary, please indicate this also in the materials and methods section.

* Please indicate the selected parameters in the automatic segmentation menu of the DAIME software (e.g. watershed %, inclusion of dark regions, etc., if indicated) in the materials and methods section.

* Please also cite the software’s original reference (H. Daims, et al.).

Minor,

(1) “In three independent experiments, the maximum diameters of the LDs were measured from two images per group …”. A spherical LD has only one diameter. The maximum/minimum diameter is relevant when the segmented objects show a non-spherical shape.

* You may remove “maximum” to avoid confusions.

(2) The graphs in Figure 8 B, F, G seem to be stretched compared to the other figures (if not a PDF conversion error).

* Please correct this, if indicated.

(3) Materials & Methods: “Peripheral regions of fat bodies were analysed using a 20x objective … ”

* Please also indicate the numerical aperture (relevant for the optical resolution) and the specifications for which optical aberrations the objective is corrected (label on the objective) in the materials and methods section.

(4) The authors discussed the alterations of the LD phenotype and occurrence of large LDs with LD growth by fusion or lipid transfer. While data of the lipid transfer between LDs is available, the fusion of the organelle e.g. due to alteration of the phospholipid profile of the monolayer is not that clear, i.e. only rarely directly monitored, yet.

* Are there any studies in insects on factors controlling LD growth by fusion available? Please comment on that.

**Reviewer #2: ** Rhodnius prolixus is an understudied organism and is associated with the disseminations of Chagas disease that has high prevalence in middle and American Countries yet is very hard to treat. Understanding the insects’ lipid metabolism with respect to lipolysis and lipophagy combined with reproductive success might become relevant to inhibit the spread of Chagas disease. The submitted manuscript describes experiments with identify if lipolytic and autophagic pathways function redundantly or synergistically using RNAi silencing experiments in R. prolixus under both - fed and fasted - conditions.

Most effects that had been reported for single-knockdowns were nicely confirmed. Most effects were not changed by dual knockdowns, as seen in the overall lack of additive effects of the individual genes in lipolysis and autophagy which is a very important finding, even though it might not be the most exciting phenotype. Interestingly however, the LD size was observed to have increased in the fat bodies of RpAtg8 + RpBmm silenced insects.

Figures are informative, presented in a very high and professional quality, thoroughly labeled and have informative figure legends. The studies have been performed, described and presented in a very solid manner and I recommend publication in PlosOne upon addressing a few minor details.

Minor:

Figure 2C, 4C: Full immunoblots of the corresponding blots should be provided in the supplementary material.

Figure 6: Due to the high error-bars, the relative mRNA-levels are presented up to 8. For better comparison possibility, please consider a representation with the ordinate value e.g. at 5 or even 3. The original Panels 6A and 6B can be shown in the supplemental material.

Figure 8: Panel C requires better explanation and labeling of the observed bands at different sizes, also which bands were used for densitometric analysis since they appear to be extremely strong and often exceed the linear range of the measurements.

Figure 8 and Fig S1: Why is the hatching frequency not normalized to 100%, Figure S1 shows 100% in dsMal. Figure S1 does not have error-bars. Please comment.

Figure 8: Some panels, e.g. 8B, 8F, look stretched in one direction only.

Figure 11 has a lower image quality compared to the others.

**Do you want your identity to be public for this peer review?** For information about this choice, including consent withdrawal, please see our Privacy Policy

Reviewer #1: No

Reviewer #2: No

---

## [Author Response · Author response to Decision Letter 1]

25 Sep 2025

Response to Reviewers - PONE-D-25-44176

Reviewer #1

In this manuscript, Santos-Araujo et al. investigate how lipolysis and lipophagy regulate lipid homeostasis in the Chagas disease related insect Rhodnius prolixus. Using RNAi technology, they downregulated RpBmm (a Brummer/ATGL ortholog) and RpAtg8 (autophagy-related) under fed and starved conditions. They measured TAG in fat body and flight muscle, studied LD morphology and well as expression of lipid genes and provided some fitness readouts. One major finding is that dual knockdown does not further increase fat-body TAG compared to single knockdowns but is accompanied by morphological alteration of LDs. These results support the notion of a tissue and state dependent coordination between lipolysis and lipophagic. They highlight the metabolic plasticity of these processes. The study is of broader interest; however, certain aspects of the TAG assay normalization and LD size quantification raise concerns.

Authors’ response: We thank the reviewer for the positive evaluation and sincerely appreciate the time and effort dedicated to carefully reading and providing constructive feedback on our manuscript. A detailed, point-by-point response is provided below, and all modifications in the revised version of the manuscript are highlighted in red.

(1) The TAG content of the fat bodies was analysed using a Kit and per fat body.

• Why was the TAG content not normalized to fat-body wet mass to control for preparation-related variability (e.g., tissue recovery), in line with previous recommendations (Tennessen et al., Methods 68, 2014, 105–115)? Please clarify.

• In this context, where insect size and/or fat body size different between groups?

Authors’ response:

We thank the reviewer for raising this important point and for indicating the excellent reference by Tennessen et al. (2014), which we now cite in the revised manuscript. We chose to express TAG values per fat body because the fat body of R. prolixus is highly lobulated and irregular, making accurate wet mass measurement difficult and prone to variability due to hemolymph retention. Another possibility would be to normalize by protein content; however, this parameter can itself be influenced by the silencing of different genes, potentially increasing, decreasing, or remaining unchanged depending on the condition. Such variability could mask effects on TAG content. For these reasons, we consider that the quantification of total TAG per fat body represents the most appropriate approach for this study. This methodology has also been adopted in previous studies using this insect model (Pontes et al., 2008; Alves-Bezerra et al., 2017; Moraes et al., 2022). Importantly, all experimental groups consisted of insects of the same age and size cohort, and no visible differences in insect size or fat body morphology were observed between treatments. We have clarified this rationale in the Materials and Methods section.

"TAG quantification was expressed as total TAG per organ, since the irregular and lobulated structure of R. prolixus fat bodies makes wet mass measurements unreliable due to hemolymph retention. Normalization by protein content was avoided, as it may vary with gene silencing and mask effects on TAG. This approach, of reporting TAG per organ, has been adopted in previous studies (Alves-Bezerra et al., 2017; Moraes et al., 2022; Pontes et al., 2008) and is consistent with recommendations by (Tennessen et al., 2014)."

(2) The authors used Nile Red labeling for image-based quantification of lipid droplet diameters. They used images of “peripheral regions” of fat bodies. Apparently, the imaged lobes can show different LD size/content (e.g. Figure 3 day 24 dsRNA). Furthermore, in the single images, it is likely that some LDs were insufficiently acquired (those partly out of focus), which could additionally limit the analysis.

• What are the criteria for the selection of the areas?

• How much of the prepared fat body area do the images approximately represent (10%, 50 %....)?

Authors’ response: We now clarify in the Methods section that peripheral regions were selected because they provide reduced tissue thickness, and better optical access for Nile Red imaging. Importantly, setting a specific region of the tissue to be observed also provides a standardization of the area analyzed across organs from different insects, ensuring comparability between samples. For each sample, two randomly chosen fields from the periphery of different lobes were analyzed. Each field corresponds to ~4–6% of the total tissue area, and thus approximately 10% of the tissue was analyzed per animal. This information has been added to the revised Methods (copied below):

"Peripheral regions of the fat body were selected for imaging because they exhibit reduced tissue thickness and allow for a standardized area of observation across organs from different insects. For each insect, two randomly chosen fields from the periphery of different lobes were acquired and analyzed. Each field represented approximately 4–6% of the total tissue area, corresponding to approximately 10% of the tissue analyzed per animal. Three independent experiments were conducted."

(3) The authors measured the diameter of LDs to highlight phenotypic differences across the different conditions. The authors cited the paper of Michele Alves-Bezerra et al. as a reference for the LD quantification process using the DAIME software, which is software to detect cells of phytoplankton, here used for LD quantification.

This paper, however, only links to the original paper (H. Daims, et al. daime, a novel image analysis program for microbial ecology and biofilm research, Environmental Microbiology 8 (2006) 200–213). In turn, in the original paper it is stated that the “edge detection” algorithm originates from the method introduced by Viles and Sieracki, 1992.

The latter evaluated different algorithms for marine plankton cell detection, not for lipid droplets. The best algorithm for the specific application was the Marr-Hildreth method with “edge strength” for automated object edge detection. Likely, this method is similar to a Laplacian of Gaussian edge detection with additional enhancement steps. Since such methods detect more abrupt contrast changes of image objects, it can be very challenging to detect and separate very closely associated LDs and putatively imaged with decreased optical resolution (20x objective).

It is not clear that this method can separate LDs in the provided images sufficiently for quantification (assuming that the overall rules of image acquisition were considered, e.g. no significant overexposed pixels, etc.).

• Please show as supplementary data the segmented image of the representative image of day 10 dsATG8 and of two other examples. These images are the basis of the quantification outcome. One can export the segmented data by right-clicking on the image thumbnail and selecting “extract object layer.” If manual refinements were necessary, please indicate this also in the materials and methods section.

Authors’ response: We appreciate and understand the reviewer’s concerns and thank him for this constructive feedback. In response, we have added representative segmented images in the Supplementary Material (Figure S1), including day 10 dsAtg8 and two additional representative conditions. The updated figure is copied below.

Figure S1. Representative segmented images of lipid droplets used for quantification. Examples of DAIME-based segmentation are shown for day 10 dsAtg8 and two additional representative experimental conditions. Lipid droplets were stained with Nile Red and segmented using the Marr-Hildreth edge detection algorithm with watershed set to 12% and exclusion of dark regions. No manual refinement was required. These images illustrate the basis for the lipid droplet diameter quantification presented in the manuscript.

• Please indicate the selected parameters in the automatic segmentation menu of the DAIME software (e.g. watershed %, inclusion of dark regions, etc., if indicated) in the materials and methods section.

Authors’ response: The parameters used in DAIME for automatic segmentation were as follows: Edge detection parameters – inclusion of dark regions: no; dark threshold: 140. General options – ignore objects up to: no. These details have now been included in the Materials and Methods section to ensure reproducibility. Text in the revised manuscript is copied here:

"The diameters of lipid droplets (LDs) were measured from two images per group using the image analysis software DAIME, following automatic edge detection–based segmentation (Alves-Bezerra et al., 2017a; Daims et al., 2006). The segmentation parameters applied in DAIME were: Edge detection – inclusion of dark regions: no; dark threshold: 140; General options – ignore objects up to: 4. Representative segmented images used for quantification are provided in Figure S1. The LD diameters were plotted in frequency histograms."

• Please also cite the software’s original reference (H. Daims, et al.).

Authors’ response: Done. The original reference for DAIME (Daims et al., 2006) was included.

(1) “In three independent experiments, the maximum diameters of the LDs were measured from two images per group …”. A spherical LD has only one diameter. The maximum/minimum diameter is relevant when the segmented objects show a non-spherical shape.

• You may remove “maximum” to avoid confusions.

Authors’ response: Done. We now refer to “diameter” only.

(2) The graphs in Figure 8 B, F, G seem to be stretched compared to the other figures (if not a PDF conversion error).

• Please correct this, if indicated.

Authors’ response: Corrected versions of the figures are now provided.

(3) Materials & Methods: “Peripheral regions of fat bodies were analysed using a 20x objective … ”

• Please also indicate the numerical aperture (relevant for the optical resolution) and the specifications for which optical aberrations the objective is corrected (label on the objective) in the materials and methods section.

Authors’ response: We have added this information to the methodology. Imaging was performed with 20x/0.7NA Plan-Apochromatic objective lens.

(4) The authors discussed the alterations of the LD phenotype and occurrence of large LDs with LD growth by fusion or lipid transfer. While data of the lipid transfer between LDs is available, the fusion of the organelle e.g. due to alteration of the phospholipid profile of the monolayer is not that clear, i.e. only rarely directly monitored, yet.

• Are there any studies in insects on factors controlling LD growth by fusion available? Please comment on that.

Authors’ response: We thank the reviewer for this suggestion. To our knowledge, direct evidence of LD fusion in insects is not available. However, work in Drosophila S2 cells has suggested that phospholipid metabolism influences LD size regulation and that phosphatidylcholine availability and phospholipid remodeling can drive LD fusion (Guo et al., 2008; Kühnlein, 2011, Krahmer et al., 2011). We have added a comment in the Discussion citing these studies. See below:

"LDs are dynamic organelles that vary in number, size, and distribution, and can grow through fusion, lipid synthesis, or lipid transfer. All these processes are influenced by metabolic signals and by LD composition (Krahmer et al., 2013; Thiam and Beller, 2017). To our knowledge, direct evidence of LD fusion in insects is not available. However, studies in Drosophila S2 cells suggest that phospholipid metabolism influences LD size regulation (Guo et al., 2008; Kühnlein, 2011), and that phosphatidylcholine availability and phospholipid remodeling can drive LD fusion (Krahmer et al., 2011)."

Authors’ Note to Reviewers – Correction in Figure 2C: Upon re-examining the original membranes of all blots, we identified an error: in Figure 2C (fat body, 10 days), the image of the actin loading control had been inadvertently misplaced. In the revised version of the manuscript, we have replaced it with the correct image.

This correction does not affect the interpretation of the data, as the actin loading control remains stable across all experiments. Nevertheless, we considered it important to make the substitution to maintain the highest level of rigor. We sincerely apologize for this mistake.

Reviewer #2

Rhodnius prolixus is an understudied organism and is associated with the disseminations of Chagas disease that has high prevalence in middle and American Countries yet is very hard to treat. Understanding the insects’ lipid metabolism with respect to lipolysis and lipophagy combined with reproductive success might become relevant to inhibit the spread of Chagas disease. The submitted manuscript describes experiments with identify if lipolytic and autophagic pathways function redundantly or synergistically using RNAi silencing experiments in R. prolixus under both - fed and fasted - conditions.

Most effects that had been reported for single-knockdowns were nicely confirmed. Most effects were not changed by dual knockdowns, as seen in the overall lack of additive effects of the individual genes in lipolysis and autophagy which is a very important finding, even though it might not be the most exciting phenotype. Interestingly however, the LD size was observed to have increased in the fat bodies of RpAtg8 + RpBmm silenced insects.

Figures are informative, presented in a very high and professional quality, thoroughly labeled and have informative figure legends. The studies have been performed, described and presented in a very solid manner and I recommend publication in PlosOne upon addressing a few minor details.

Authors’ response: We thank the reviewer for the positive evaluation and sincerely appreciate the time and effort dedicated to carefully reading and providing constructive feedback on our manuscript. A detailed, point-by-point response is provided below, and all modifications in the revised version of the manuscript are highlighted in red.

Minor:

Figure 2C, 4C: Full immunoblots of the corresponding blots should be provided in the supplementary material.

Authors’ response: We have included the uncropped full blots in the Supplementary Material (Figure S2). Also copied below:

Figure S2. Full, uncropped immunoblots corresponding to Figures 2C and 4C. Uncropped images of the original blots are shown to document the full experimental results. The specific regions used in the main figures are indicated in red.

Figure 6: Due to the high error-bars, the relative mRNA-levels are presented up to 8. For better comparison possibility, please consider a representation with the ordinate value e.g. at 5 or even 3. The original Panels 6A and 6B can be shown in the supplemental material.

Authors’ response: We agree with the reviewer. In the revised manuscript, the main graphs are now presented with a y-axis divided into two segments, with the lower segment limited to 3 for improved clarity. The original full-scale graphs are provided in the Supplementary Material as Fig. S3. The revised Fig6 is copied below.

Figure 8: Panel C requires better explanation and labeling of the observed bands at different sizes, also which bands were used for densitometric analysis since they appear to be extremely strong and often exceed the linear range of the measurements.

Authors’ response: We apologize for not indicating the Vg bands in the original version. The fragments corresponding to Vg subunits have been previously characterized (Masuda and Oliveira, 1985). In the revised figure, we have added arrowheads to identify the bands representing Vg, and the figure legend has been corrected accordingly. Those are the bands that were used for densitometric analysis. We have also clarified this point in the description of the figure within the Results section. The following text was edited/added to the revised manuscript:

"Given its pivotal role in reproduction, Vg levels in the hemolymph were analyzed by SDS-PAGE. The fragments representing Vg subunits have been previously characterized (Masuda and Oliveira, 1985). Both the representative gel (Figure 8C, arrowheads point to Vg subunits) and the

---

## [Decision Letter · Decision Letter 1]

7 Oct 2025

Dear Dr. Ramos,

We look forward to receiving your revised manuscript.

Kind regards,

Monika Oberer

Academic Editor

PLOS ONE

Journal Requirements:

Additional Editor Comments:

Dear Authors,

Your manuscript has been significantly improved. One reviewer has specific questions (and specific instructions) in Q3 that should be carried out before final acceptance of your manuscript. Please implement this data analysis in your revision.

Sincerly,

Monika Oberer

Reviewers' comments:

Reviewer's Responses to Questions

**Comments to the Author**

Reviewer #1: (No Response)

2. Is the manuscript technically sound, and do the data support the conclusions?

Reviewer #1: Yes

3. Has the statistical analysis been performed appropriately and rigorously?

Reviewer #1: Yes

4. Have the authors made all data underlying the findings in their manuscript fully available?

Reviewer #1: No

5. Is the manuscript presented in an intelligible fashion and written in standard English?

Reviewer #1: Yes

Reviewer #1: The authors addressed most of my questions satisfactorily, except for question 3. The images in Fig. S1 R1 do not show the segmentation output (binary; black-and-white); instead, they are still grayscale images and apparently do not represent the segmentation data produced by the software. To access the segmentation, one must extract it from the segmentation data layer (this is a fast, simple process in DAIME 3: Import the data in DAIME. After segmentation, select the imported original image. Right-click and choose “Extract object layer”. The segmentation layer appears below. Then right-click the segmentation layer and select “Show in Visualizer”. In the Visualizer, click the camera icon to take a snapshot and save it.

The segmented images are the basis for the quantification of the LDs (diameters, histograms). Based on these data, one can evaluate the efficiency of LD detection and, particularly, how closely associated and more diffuse LDs were separated. Since reliable image-based LD detection can be challenging and is used in different fields, demonstrating the efficiency of DAIME for single-LD detection/separation in such images would be valuable. Furthermore, these images serve as a control for the quantification process and related outcome. Please provide the segmentation data.

**Do you want your identity to be public for this peer review?** For information about this choice, including consent withdrawal, please see our Privacy Policy

Reviewer #1: No

---

## [Author Response · Author response to Decision Letter 2]

21 Oct 2025

Response to Reviewers - PONE-D-25-44176 - R2

Reviewer #1

Reviewer #1: The authors addressed most of my questions satisfactorily, except for question 3. The images in Fig. S1 R1 do not show the segmentation output (binary; black-and-white); instead, they are still grayscale images and apparently do not represent the segmentation data produced by the software. To access the segmentation, one must extract it from the segmentation data layer (this is a fast, simple process in DAIME 3: Import the data in DAIME. After segmentation, select the imported original image. Right-click and choose “Extract object layer”. The segmentation layer appears below. Then right-click the segmentation layer and select “Show in Visualizer”. In the Visualizer, click the camera icon to take a snapshot and save it.

The segmented images are the basis for the quantification of the LDs (diameters, histograms). Based on these data, one can evaluate the efficiency of LD detection and, particularly, how closely associated and more diffuse LDs were separated. Since reliable image-based LD detection can be challenging and is used in different fields, demonstrating the efficiency of DAIME for single-LD detection/separation in such images would be valuable. Furthermore, these images serve as a control for the quantification process and related outcome. Please provide the segmentation data.

Authors’ response: We sincerely apologize for the confusion caused in our previous revision. In our first response, we mistakenly submitted grayscale (black-and-white) images instead of the actual segmented layer images extracted from the DAIME software. We would like to emphasize that this was purely a submission error and not an intentional omission or withholding of data. We fully acknowledge the reviewer’s concern and understand the importance of presenting the correct segmentation outputs that correspond to the original images, as these are essential for validating the accuracy of LD quantification.

We have now corrected this oversight and are providing, for your evaluation, the segmented layer images corresponding to four representative examples from each of the time points analyzed in Atg8-silenced fat bodies. These images were properly extracted from the segmentation layer in DAIME and directly represent the datasets used for LD quantification.

Figure S1 was revised to show the actual segmentation layers of three representative images.

We thank the reviewer for the careful and constructive revision.

---

## [Decision Letter · Decision Letter 2]

26 Oct 2025

Dual silencing of lipophagy and lipolysis in Rhodnius prolixus induces lipid droplet remodeling without TAG accumulation in the fat body

PONE-D-25-44176R2

Dear Dr. Ramos,

We’re pleased to inform you that your manuscript has been judged scientifically suitable for publication and will be formally accepted for publication once it meets all outstanding technical requirements.

Kind regards,

Monika Oberer

Academic Editor

PLOS ONE

Additional Editor Comments (optional):

Reviewers' comments:

Reviewer's Responses to Questions

**Comments to the Author**

Reviewer #1: All comments have been addressed

2. Is the manuscript technically sound, and do the data support the conclusions?

Reviewer #1: Yes

3. Has the statistical analysis been performed appropriately and rigorously?

Reviewer #1: Yes

4. Have the authors made all data underlying the findings in their manuscript fully available?

Reviewer #1: Yes

5. Is the manuscript presented in an intelligible fashion and written in standard English?

Reviewer #1: Yes

Reviewer #1: (No Response)

**Do you want your identity to be public for this peer review?** For information about this choice, including consent withdrawal, please see our Privacy Policy

Reviewer #1: No

---

## [Editor Report · Acceptance letter]

PONE-D-25-44176R2

PLOS ONE

Dear Dr. Ramos,

I'm pleased to inform you that your manuscript has been deemed suitable for publication in PLOS ONE. Congratulations! Your manuscript is now being handed over to our production team.

Kind regards,

on behalf of

Dr. Monika Oberer

Academic Editor

PLOS ONE